# DD-Ranking: Rethinking the Evaluation of Dataset Distillation

## Abstract

Dataset distillation has provided a reliable solution for data compression, where models trained on the resulting smaller synthetic datasets achieve performance comparable to those trained on the original datasets. To further improve the performance of synthetic datasets, various training pipelines and optimization objectives have been proposed, greatly advancing the field of dataset distillation. Recent decoupled dataset distillation methods introduce soft labels and stronger data augmentation during the post-evaluation phase and scale dataset distillation up to larger datasets. However, this raises a question: *Is accuracy still a reliable metric to fairly evaluate dataset distillation methods?* Our empirical findings suggest that the performance improvements of these methods often stem from additional techniques rather than the inherent quality of the images themselves, with even randomly sampled images achieving superior results. Such misaligned evaluation settings severely hinder the development of DD. Therefore, we propose DD-Ranking, a unified evaluation framework, along with new general evaluation metrics to uncover the true performance improvements achieved by different methods. By refocusing on the actual information enhancement of distilled dataset, DD-Ranking provides a more comprehensive and fair evaluation standard for future research advancements.

## 1 Introduction

With the rapid advancement of deep learning, training increasingly complex and more complex models on large scale datasets has become a standard paradigm, achieving remarkable performance in various fields, such as computer vision (Dosovitskiy et al., 2020; He et al., 2016) and natural language processing (Brown et al., 2020; Devlin et al., 2018). However, this process often incurs substantial computational and storage demands, significantly hindering deployment across diverse scenarios. Dataset distillation (DD) (Wang et al., 2018), as a recent promising solution for dataset compression, offers novel insights to address these challenges. In recent years, diverse training pipelines (Deng and Russakovsky, 2022; Yin et al., 2024; Zhao and Bilen, 2022a) and optimization objectives (Cazenavette et al., 2022; Zhao and Bilen, 2021a; Zhao et al., 2021) have been proposed, driving rapid advancement in the field of dataset distillation.

To further enhance the testing accuracy of models trained on synthetic datasets during the post-evaluation phase, recent studies have incorporated general performance boosting techniques (e.g., soft labels) into the evaluation process. Some methods jointly optimize the generated images and their corresponding unique soft labels (Guo et al., 2023; Loo et al., 2022b), while decoupled dataset distillation methods (Shao et al., 2024a; Su et al., 2024b; Sun et al., 2024; Yin et al., 2024) utilize epoch-wise soft labels provided by pre-trained teacher models during post-evaluation phase. Although these works successfully demonstrate that soft labels significantly improve testing accuracy of the validation models, their soft label implementation strategies differ substantially, and performance comparisons with prior methods often fail to account for gains attributable to soft labels.

Furthermore, subsequent studies frequently employ more intensive data augmentation, superior optimizers, and refined training hyper-parameters (Cui et al., 2025; Shao et al., 2024c) during evaluation to maximize model performance, with even randomly sampled images achieving superior results under better post-evaluation settings. This practice conflates genuine improvements in dataset quality with performance variations caused by inconsistent evaluation settings, severely impeding progress in dataset distillation and directing subsequent improvements toward suboptimal directions.

| Config | DC | DSA | DM | MTT | DataDAM | DATM | SRe2L | RDED | CDA | DWA | D4M | EDC | G-VBSM |
|---|---|---|---|---|---|---|---|---|---|---|---|---|---|
| Epoch | 1K | 1K | 1K | 1K | 1K | 1K | 300 | 300 | 300 | 300 | 300 | 300 | 300 |
| Batch Size | 256 | 256 | 256 | 256 | 256 | 256 | 1024 | 100 | 128 | 128 | 1024 | 100 | 1024 |
| Optimizer | SGD | SGD | SGD | SGD | SGD | SGD | AdamW | AdamW | AdamW | AdamW | AdamW | AdamW | AdamW |
| LR Scheduler | step | step | step | step | step | step | cosine | cosine | cosine | cosine | cosine | cosine | cosine |
| Label Type | hard | hard | hard | hard | hard | soft | soft | soft | soft | soft | soft | soft | soft |
| Soft Label | - | - | - | - | - | single | multiple | multiple | multiple | multiple | multiple | multiple | multiple |
| Loss Function | CE | CE | CE | CE | CE | SCE | KL | KL | KL | KL | KL | MSE | MSE |
| Teacher Model | - | - | - | - | - | single | single | single | single | single | single | ensemble | ensemble |
| DSA | No | Yes | Yes | Yes | Yes | Yes | No | No | No | No | No | No | No |
| ZCA | No | No | No | Yes | No | Yes | No | No | No | No | No | No | No |
| ResizeCrop | No | No | No | No | No | No | Yes | Yes | Yes | Yes | Yes | Yes | Yes |
| CropRange | - | - | - | - | - | - | 0.08, 1.0 | 0.5, 1.0 | 0.08, 1.0 | 0.08, 1.0 | 0.08, 1.0 | 0.5, 1.0 | 0.08, 1.0 |
| PatchShuffle | No | No | No | No | No | No | No | Yes | No | No | No | Yes | No |
| CutMix | No | No | No | No | No | No | Yes | Yes | Yes | Yes | Yes | Yes | Yes |

Table 1: Evaluation configurations of various dataset distillation methods. We separate agent model training hyperparameters (top) and data augmentation (bottom). For each row, different colors highlight the differences in the evaluation setting.

Based on the aforementioned discussion, we must emphasize that in the growing field of dataset distillation, relying solely on the testing accuracy of validation model as the exclusive criterion for assessing the quality of synthetic datasets exhibits significant unreliability and unfairness when applied across varying settings.

To address these issues, we propose DD-Ranking, a unified evaluation framework, and introduce a new fair and generalizable metric to realign with the original objectives of dataset distillation. Specifically, we first test evaluation models using randomly sampled images under the evaluation settings of various distillation methods to establish baseline performance for different settings. The performance of generated images is then calibrated by calculating the difference from this baseline. On the other hand, we compute the difference between the performance of synthetic datasets under the hard label settings and the maximum achievable performance using the full original dataset. By jointly applying these two adaptive metrics to evaluate existing distillation methods, we derive a new performance indicator that reveals the true differences in distillation capabilities among methods. Building upon this, we also propose a novel metric for evaluating data augmentations. We further examine the robustness of the introduced metrics across diverse application scenarios.

DD-Ranking addresses the inconsistencies present in existing dataset distillation evaluation protocols and unifies various methods under a fair and standardized evaluation framework, thereby establishing a solid baseline and offering valuable insights for future research. The contributions of our benchmark are threefold. First, we standardize evaluation metrics for dataset distillation, resolving the persistent issue of unfair comparisons in test accuracy across different methods. Second, experimental observations from DD-Ranking demonstrate that previous performance improvements commonly originate from the enhanced model training techniques instead of the distilled dataset. Thus, DD-Ranking encourages the community to direct future efforts toward enhancing the informativeness of synthetic data. Third, building upon the era of dataset distillation, we introduce a general and robust metric that serves as a novel evaluation criterion, with broader applicability across data-centric AI tasks.

## 2 MOTIVATION

### 2.1 OVERVIEW OF UNFAIRNESS

The conventional approach to evaluating dataset distillation methods relies on measuring the **test accuracy** of a classification model trained on the distilled dataset[1]. However, we have identified substantial unfairness in this evaluation paradigm stemming from highly inconsistent training configurations for the classification model. Table 1 presents a comparative analysis of training parameters and data augmentation employed by various dataset distillation methods (Cazenavette et al., 2022; Guo et al., 2023; Sajedi et al., 2023; Shao et al., 2024a;c; Su et al., 2024a; Sun et al., 2024; Yin and Shen, 2024; Yin et al., 2024; Zhao and Bilen, 2021b; 2023; Zhao et al., 2021) on the same

---

[1]Our discussion focuses exclusively on image classification datasets, as these are most frequently used.

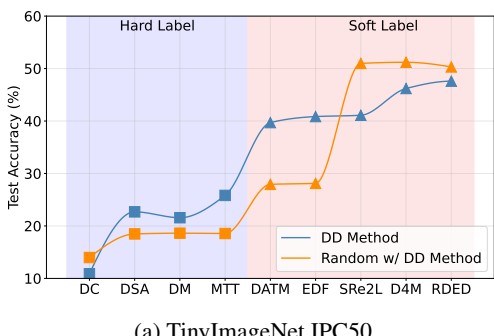
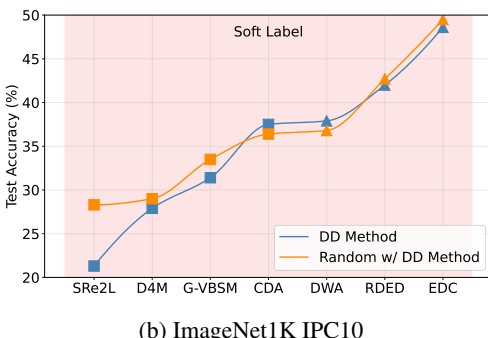

(a) TinyImageNet IPC50      (b) ImageNet1K IPC10

Figure 1: Test accuracies of the model trained on synthetic data distilled by various DD methods and on randomly selected data. Despite soft labels being able to significantly improve the test accuracy, DD methods may fail to outperform random selection under the same training setting.

target dataset. We use different colors to highlight the differences in the current dataset distillation evaluation settings. We believe that the performance evaluated without a unified and standardized benchmark is not reliable for a fair comparison. Among these inconsistencies, two critical factors significantly undermine the fairness of current evaluation protocols: label representation (including the corresponding loss function) and data augmentation techniques.

## 2.2 SOFT LABELS

**Soft labels significantly improve the test accuracy.** Soft labels, which represent probability distributions rather than discrete values, are widely used in knowledge distillation and increasingly in dataset distillation evaluation. Here, distilled images are paired with soft labels from a pretrained teacher. For example, DATM (Guo et al., 2023) optimizes labels concurrently with data, while SRe2L (Yin et al., 2024) generates them at test time. This minimizes the loss (e.g., KL divergence) against the soft labels, leading to consistently higher performance metrics (Figure 1).

| batch size | 128 | 1024 |
|---|---|---|
| ImageNet-1K | 0.5 | 0.3 |

| noise | $\sigma = 0.05$ | $\sigma = 0.5$ |
|---|---|---|
| TinyImageNet | 49.9 (↑ 26.0) | 14.6 (↑ 3.9) |
| ImageNet | 60.6 (↑ 24.9) | 6.8 (↑ 5.6) |

| Method | EDC | | RDED | |
|---|---|---|---|---|
| IPC | 10 | 50 | 10 | 50 |
| w/ aug. | 48.6 | 58.0 | 42.0 | 56.5 |
| w/o aug. | 12.5 | 39.7 | 15.3 | 27.9 |

(a) The performances of random noise outperform the random guessing (0.1) with soft labels.

(b) Noised images under different noise levels with soft labels significantly outperform hard label performance (improvements are shown in the bracket).

(c) Data augmentation largely contributes to the high accuracy, especially on high-resolution datasets.

Table 2: (a) Improvements of soft labels on random noises with different standard deviations. (b) Ablation of data augmentation on ImageNet1K.

**Improvements originate from knowledge distillation, instead of synthetic data.** We argue that the accuracy gains are predominantly attributable to knowledge distillation from soft labels rather than the intrinsic informativeness of the distilled data. We substantiated this by comparing baselines against random noise and randomly selected samples, both annotated with soft labels. Crucially, we kept all training parameters—such as the teacher model, learning rate, and optimizer—identical across comparisons. As demonstrated in Figure 1, random data with soft labels consistently outperforms baseline-distilled data, echoing Qin et al. (2024). Furthermore, even random noise patterns with soft labels can achieve non-negligible accuracy well above random guessing (Table 2a). Also, images disturbed by high noise levels with soft labels still outperform their hard label performance significantly (Table 2b). These findings indicate that while soft labels boost metrics, they mask the true informativeness of the distilled data.

## 2.3 DATA AUGMENTATION

Data augmentation is a widely used technique to enhance model training performance. Current dataset distillation methods also apply various augmentation techniques during their evaluation process. As shown in Table 1, there is significant diversity in the augmentation strategies used by existing dataset distillation methods, with different approaches typically adopting different sets of transformations. However, this variation makes it difficult to fairly evaluate and compare different dataset distillation methods because improvements in test accuracy brought about by data augmentation do not necessarily reflect the inherent quality of the distilled data itself.

To better demonstrate this claim, we conducted a comparative analysis of two established baseline methods, measuring their performance both with and without their respective data augmentation. As depicted in Table 2c, a substantial portion of the reported performance gains can be directly attributed to augmentation rather than to the intrinsic quality of the distilled datasets. Therefore, similar to soft labels, these results highlight the need for new evaluation metrics that more accurately capture the true informational value of distilled data, instead of relying solely on raw test accuracy that can be inflated by augmentation techniques.

## 3 DD-RANKING

### 3.1 OVERVIEW

Motivated by the unfairness above, we introduce DD-Ranking. DD-Ranking is an integrated and easy-to-use evaluation benchmark for dataset distillation (DD). It aims to provide a fair evaluation scheme for DD methods that can decouple the impacts from knowledge distillation and data augmentation to reflect the real informativeness of the distilled data. Under the finding that the test accuracy no longer fits the need for fair and comprehensive evaluation, we design new metrics for both the label representation and data augmentation.

### 3.2 LABEL-ROBUST SCORE

**Hard label recovery.** The initial goal of dataset distillation is to synthesize a small number of data points that do not need to come from the correct data distribution, but will, when given to the learning algorithm as training data, approximate the model trained on the original data (Wang et al., 2018). Given that almost all existing classification datasets use hard label annotation, we think it is crucial for DD methods to maintain good performance with hard labels. To this end, we propose the **hard label recovery (HLR)**. Specifically, for both hard-label-based and soft-label-based methods, we evaluate the test accuracy of the distilled data and that of the original dataset with hard labeling, denoted as $\text{acc}_{\text{syn-hard}}$ and $\text{acc}_{\text{real-hard}}$, respectively. The hard label recovery is computed by taking the difference:

$$\text{HLR} = \text{acc}_{\text{real-hard}} - \text{acc}_{\text{syn-hard}} \tag{1}$$

A smaller HLR indicates that the distilled data enables the model to recover more of the performance of the same model trained on the full dataset.

**Improvement over random.** Despite the popularity of soft labels, it's not fair to directly compare methods with soft labels against those with hard labels. Also, there isn't a unified recipe for soft-label-based evaluation, and differences such as soft labels per sample, loss function, and temperature could significantly impact the results. This makes it difficult to compare different soft-label-based methods. To make different methods comparable under mixed label types, we propose **improvement over random (IOR)**. This metric is based on the common sense that any DD method should at least outperform random selection under the same evaluation recipe, and we use the relative performance improvements over random selection to compare any pair of DD methods. Specifically, denote the test accuracys of the model trained on synthetic data and a same-sized random subset with any label type as $\text{acc}_{\text{syn-any}}$ and $\text{acc}_{\text{rdm-any}}$, respectively. For each DD method, we keep the model training settings (such as data augmentation, loss function, learning rate, etc.) the same on synthetic and random data. Then, the IOR is computed by:

$$\text{IOR} = \text{acc}_{\text{syn-any}} - \text{acc}_{\text{rdm-any}} \tag{2}$$

IOR is positively related to the performance of DD methods. By doing so, we can effectively disentangle the improvement brought solely by knowledge distillation and reflect the true informativeness.

**Label-robust score** Combining hard label recovery (HLR) and improvement over random (IOR), we present the label-robust score (LRS). LRS first takes a weighted sum $\alpha$ of HLR and IOR via a weight parameter $\lambda$ as follows:

$$\alpha = \lambda\text{IOR} - (1 - \lambda)\text{HLR} \tag{3}$$

We assign a negative mark to HLR so that both parts of the sum are positively correlated with the performance. The raw range of $\alpha$ is between $[-1, 1]$, so we normalize LRS to the range $[0, 1]$ by letting $\text{LRS} = 100\% \times (e^\alpha - e^{-1})/(e - e^{-1})$. A higher LRS indicates that the distilled dataset of the corresponding method is more robust to the label representation and has richer information.

## 3.3 AUGMENTATION-ROBUST SCORE

Data augmentation, as a trick to enhance model training, doesn't reveal the quality of the dataset itself. Thus, the improvement in test accuracy brought merely by data augmentation at test time should not be attributed to the effectiveness of the dataset distillation method. To disentangle data augmentation's impact, we introduce the **augmentation-robust score (ARS)** which continues to leverage the relative improvement over randomly selected data. Specifically, we first evaluate synthetic data and a randomly selected subset under the same setting to obtain $\text{acc}_{\text{syn-aug}}$ and $\text{acc}_{\text{rdm-aug}}$ (same as IOR). Next, we evaluate both synthetic data and random data again without the data augmentation, and results are denoted as $\text{acc}_{\text{syn-naug}}$ and $\text{acc}_{\text{rdm-naug}}$.

We claim that an informative subset via distillation should surpass any randomly selected subset of the same size, regardless of the use of data augmentation. Thus, both differences, $\text{acc}_{\text{syn-aug}} - \text{acc}_{\text{rdm-aug}}$ and $\text{acc}_{\text{syn-naug}} - \text{acc}_{\text{rdm-naug}}$, are positively correlated to the real informativeness of the distilled dataset. We take a weighted sum of the two differences

$$\beta = \gamma(\text{acc}_{\text{syn-aug}} - \text{acc}_{\text{rdm-aug}}) + (1 - \gamma)(\text{acc}_{\text{syn-naug}} - \text{acc}_{\text{rdm-naug}}) \tag{4}$$

and use a similar normalization method to compute ARS. A higher ARS indicates that the distilled dataset of the corresponding method is more robust to data augmentation.

## 3.4 THEORETICAL FOUNDATION

We provide a theoretical analysis demonstrating why LRS enables fair comparison between methods with different settings (ARS follows analogously).

**Definition of fair comparison** A fair comparison in DD isolates data quality contributions while neutralizing setting-dependent performance gains. Formally, let $q_A, q_B$ denote the intrinsic data quality of methods A and B, and $\beta(\pi)$ represent performance boost from evaluation setting $\pi$ (independent of data quality). The comparison is fair if and only if it directly compares $q_A, q_B$.

**Proof of LRS's fairness** Consider method A distilling $\mathcal{D}_A$ with protocol $\pi_A$ (e.g., hard labels, batch size 256, step learning rate scheduler) and method B distilling $\mathcal{D}_B$ with protocol $\pi_B$ (e.g., soft labels, batch size 1024, cosine learning rate scheduler). We decompose performance as

$$\text{Acc}(\mathcal{D}_A, \pi_A) = q_A + \beta(\pi_A) + \epsilon_A, \quad \text{Acc}(\mathcal{D}_B, \pi_B) = q_B + \beta(\pi_B) + \epsilon_B \tag{5}$$

where $\epsilon \sim \mathcal{N}(0, 1)$ is the residual term. Direct accuracy comparison is unfair due to unknown relationship between $\beta(\pi_A)$ and $\beta(\pi_B)$. Considering random selection baseline

$$\text{Acc}(\mathcal{D}_{\text{r}}, \pi_{\text{r}}) = q_{\text{r}} + \beta(\pi_{\text{r}}) + \epsilon_{\text{r}} \tag{6}$$

we can show IOR's fairness (HLR follows similarly) by:

$$\begin{aligned}
\text{IOR}_A = \text{Acc}(\mathcal{D}_A, \pi_A) - \text{Acc}(\mathcal{D}_{\text{r}_A}, \pi_{\text{r}_A}) = q_A - q_{\text{r}_A} + \epsilon_A - \epsilon_{\text{r}_A} \\
\text{IOR}_B = \text{Acc}(\mathcal{D}_B, \pi_B) - \text{Acc}(\mathcal{D}_{\text{r}_B}, \pi_{\text{r}_B}) = q_B - q_{\text{r}_B} + \epsilon_B - \epsilon_{\text{r}_B}
\end{aligned} \tag{7}$$

where $\beta$ terms cancel since $\pi_A = \pi_{\text{r}_A}$ and $\pi_B = \pi_{\text{r}_B}$. Thus $\mathbb{E}[\text{IOR}_A - \text{IOR}_B] = \mathbb{E}[q_A - q_B] - \mathbb{E}[q_{\text{rand}_A} - q_{\text{rand}_B}] + \mathbb{E}[\epsilon_A - \epsilon_B] - \mathbb{E}[\epsilon_{\text{rand}_A} - \epsilon_{\text{rand}_B}]$. Since the expected intrinsic quality of random selection is constant for the same original dataset $\mathcal{D}$, we have $\mathbb{E}[q_{\text{rand}_A} - q_{\text{rand}_B}] = 0$ and $\mathbb{E}[\epsilon] = 0$, yielding $\mathbb{E}[\text{IOR}_A - \text{IOR}_B] = \mathbb{E}[q_A - q_B]$. This demonstrates that IOR comparison directly measures intrinsic synthetic data quality without data-irrelevant factors.

| IPC | 1 | | | 10 | | | 50 | | |
| --- | --- | --- | --- | --- | --- | --- | --- | --- | --- |
| Metric | HLR↓ | IOR↑ | LRS↑ | HLR↓ | IOR↑ | LRS↑ | HLR↓ | IOR↑ | LRS↑ |
| DC | 52.7 | 12.4 | 19.1 | 36.7 | 18.5 | 23.2 | 26.3 | 12.3 | 24.0 |
| DSA | 58.9 | 13.2 | 18.2 | 35.1 | 19.6 | 23.7 | 27.4 | 11.0 | 23.5 |
| MTT | 42.2 | 27.6 | 23.9 | **23.7** | 30.9 | 28.4 | **16.5** | 20.5 | 27.8 |
| DM | 61.4 | 8.7 | 17.0 | 39.4 | 16.1 | 22.2 | 25.1 | 12.7 | 24.3 |
| DATADAM | 49.9 | 15.6 | 20.0 | 34.8 | 19.9 | 23.8 | 21.9 | 15.8 | 25.6 |
| DATM | **41.9** | **30.8** | **24.6** | 26.8 | **35.1** | **28.7** | 18.9 | **23.9** | **28.0** |
| SRe2L | 69.9 | -0.3 | 14.3 | 67.8 | -5.7 | 13.8 | 62.9 | -6.5 | 14.4 |
| RDED | 60.6 | 2.4 | 16.2 | 50.7 | 1.1 | 17.6 | 36.0 | -1.6 | 19.6 |
| D4M | 51.1 | 6.7 | 18.4 | 39.9 | 9.1 | 20.8 | 27.0 | 6.6 | 22.8 |

Table 3: Label-robust score evaluation results on CIFAR-10. We also report the hard-label recovery and improvement over random for a more comprehensive comparison. The color scheme corresponds to that of Figure 1. The $\lambda$ is set to 0.5 for this and the following results. On CIFAR-10, hard-label-based methods perform generally better.

# 4 RESULTS

## 4.1 EVALUATION SETTINGS

**Baseline.** We evaluate a wide range of representative works in dataset distillation. For hard-label methods, we evaluate DC (Zhao et al., 2021), DSA (Zhao and Bilen, 2021b), MTT (Cazenavette et al., 2022), DM (Zhao and Bilen, 2021a), and DataDAM (Sajedi et al., 2023). For soft-label methods, we evaluate SRe2L (Yin et al., 2024), DATM (Guo et al., 2023), EDF (Wang et al., 2025a), DWA (Du et al., 2024a), RDED (Sun et al., 2024), CDA (Yin and Shen, 2024), EDC (Shao et al., 2024c), and G-VBSM (Shao et al., 2024b). In the case where the method provides its distilled data, we adopt it directly. In the case where the distilled data is absent, we strictly follow their implementation provided in both the paper and code repo to replicate their results.

**Dataset.** We report DD-Ranking benchmarking results on the four existing datasets: CIFAR-10 (Krizhevsky, 2009), CIFAR-100 (Krizhevsky, 2009), TinyImageNet (Le and Yang, 2015), and ImageNet1K (Russakovsky et al., 2015). The resolution of images in CIFAR-10 and CIFAR-100 is $32 \times 32$. The resolution of images in TinyImageNet is $64 \times 64$. The resolution of images in ImageNet1K is $224 \times 224$.

**Model.** For each baseline method, we use the reported model for evaluation. This includes ConvNet of depth 3 and 4 with instance normalization, ConvNet of depth 3 and 4 with batch normalization, and ResNet-18 (He et al., 2015). We also incorporate AlexNet (Krizhevsky et al., 2012), ResNet-50, VGG-11 (Simonyan and Zisserman, 2015), Swin-T-tiny (Liu et al., 2021), and ViT-base (Dosovitskiy et al., 2021) to validate the robustness of DD-Ranking on different architectures.

**DD-Ranking evaluation.** The evaluation is performed **5 times** with different random seeds. We report **the mean value** in the following tables. Standard deviations are presented in the Appedix. When computing the accuracy under hard labels, we perform the hyperparameter search for the learning rate and report the best one. When computing the accuracy under soft labels, we regard the learning provided by each method as the **optimal learning rate** by default, and the learning rate search is performed for random selection.

## 4.2 LABEL-ROBUST SCORE

**Results on CIFAR-10, CIFAR-100, and TinyImageNet.** Tables 3, 4, and 5 present LRS evaluation results on CIFAR-10, CIFAR-100, and TinyImageNet, respectively. Among hard-label-based methods, trajectory matching (MTT) achieves the best performance, outperforming both gradient matching approaches (DC and DSA) and distribution matching methods (DM and DataDAM). As IPC increases, the distribution matching methods perform better than the gradient matching methods. Within the soft-label-based category, methods that optimize one-to-one soft labels jointly with synthetic data (DATM) demonstrate superior performance compared to approaches that directly utilize soft labels

| IPC | 1 | | | 10 | | | 50 | | |
|-----|------|------|------|------|------|------|------|------|------|
| Metric | HLR↓ | IOR↑ | LRS↑ | HLR↓ | IOR↑ | LRS↑ | HLR↓ | IOR↑ | LRS↑ |
| DC | 39.4 | 8.4 | 20.8 | 25.5 | 12.7 | 24.2 | 21.8 | 1.1 | 22.7 |
| DSA | 46.0 | 8.5 | 19.6 | 26.1 | 13.5 | 24.3 | 21.2 | 2.0 | 23.0 |
| MTT | 35.2 | 16.7 | 23.1 | **18.0** | **20.7** | **27.5** | 12.1 | 11.6 | 26.8 |
| DM | 48.0 | 6.1 | 18.9 | 30.1 | 10.7 | 23.0 | 16.6 | 7.2 | 24.9 |
| DATADAM | 45.2 | 9.1 | 19.9 | 25.9 | 14.8 | 24.6 | 12.4 | 11.8 | 26.8 |
| DATM | **24.1** | **18.5** | **25.7** | 18.9 | 18.4 | 26.8 | **10.3** | **26.1** | **30.4** |
| SRe2L | 52.7 | -1.9 | 16.7 | 50.5 | -14.8 | 15.0 | 46.2 | -11.5 | 16.2 |
| RDED | 45.6 | -0.5 | 18.1 | 37.5 | -1.2 | 19.4 | 27.3 | -1.5 | 21.2 |
| D4M | 30.9 | 10.0 | 22.7 | 40.1 | 9.7 | 20.9 | 26.7 | 13.5 | 24.2 |

Table 4: LRS, HLR, and IOR evaluation results on CIFAR-100. DATM constantly performs the best and outperforms random selection to a large extent. This implies that soft labels are effective in improving synthetic data when used properly.

| IPC | 1 | | | 10 | | | 50 | | |
|-----|------|------|------|------|------|------|------|------|------|
| Metric | HLR↓ | IOR↑ | LRS↑ | HLR↓ | IOR↑ | LRS↑ | HLR↓ | IOR↑ | LRS↑ |
| DC | 28.6 | 3.9 | 22.0 | 21.5 | 7.1 | 23.9 | 21.3 | -2.1 | 22.2 |
| DSA | 30.3 | 3.7 | 21.6 | 20.3 | 6.8 | 24.1 | 17.6 | 7.6 | 25.8 |
| MTT | 30.7 | 5.8 | 21.9 | **15.6** | 14.6 | **26.7** | 15.6 | 10.2 | 26.4 |
| DM | 36.7 | 2.3 | 20.2 | 26.2 | 7.5 | 23.1 | 18.9 | 5.3 | 24.1 |
| DATM | **25.4** | 8.6 | **23.5** | 18.3 | 14.2 | 26.0 | **13.5** | 15.1 | 27.2 |
| EDF | 25.8 | **9.2** | **23.5** | 18.5 | **15.4** | 26.2 | 13.8 | 15.9 | **27.3** |
| SRe2L | 45.6 | -1.8 | 15.4 | 43.6 | -8.5 | 17.1 | 33.6 | -9.6 | 18.6 |
| RDED | 34.0 | 3.9 | 21.0 | 25.6 | 1.8 | 23.7 | 15.2 | -0.6 | 23.7 |
| D4M | 40.6 | -3.0 | 18.6 | 35.6 | -5.8 | 18.9 | 27.7 | 12.8 | 23.8 |

Table 5: LRS, HLR, and IOR evaluation results on TinyImageNet. For decoupled methods, D4M appears to be more effective when IPC is large, and RDED performs better at smaller IPCs.

from teacher models (D4M, SRe2L, and RDED). D4M, a generative modeling approach, outperforms decoupled methods as IPC increases. Across all methods, DATM emerges as the strongest baseline. Notably, hard-label-based methods yield results closer to full-dataset performance with hard labels and exhibit greater improvement over random data selection compared to their soft-label counterparts.

**Results on ImageNet1K.** Table 6 presents LRS results of various methods on ImageNet1K. All existing methods capable of efficiently scaling to ImageNet1K employ soft labeling techniques. Remarkably, current DD methods consistently underperform random selection across most IPC settings when soft labeling is also applied to randomly selected data. This performance gap widens as IPC increases. While these methods achieve high accuracy when using soft labels, their performance under hard labels deteriorates significantly, revealing a substantial gap compared to the real dataset.

**Findings.** Based on these results, we identify three key insights.

*i) Test accuracy is not a reliable metric when soft labels are employed.* Soft labels demonstrate even higher effectiveness on random data. Notably, on TinyImageNet and ImageNet1K, classifiers trained on random data with soft labels consistently **outperform** those trained on DD-synthesized data. While DATM maintains an advantage over random selection on TinyImageNet, this improvement diminishes substantially when soft labels are applied to random data. This observation reinforces our claim that accuracy improvements with soft labels primarily stem from knowledge distillation rather than the intrinsic informativeness of synthetic data.

*ii) Soft labels enhance synthetic dataset informativeness when jointly optimized.* Among soft-label-based methods, DATM and EDF employ a distinct approach by assigning unique soft labels to each sample and jointly optimizing both samples and labels during distillation. Unlike generative and decoupled methods that generate soft labels at test time, these optimized soft labels improve synthetic

| IPC | 1 | | | 10 | | | 50 | | |
|---|---|---|---|---|---|---|---|---|---|
| Metric | HLR↓ | IOR↑ | LRS↑ | HLR↓ | IOR↑ | LRS↑ | HLR↓ | IOR↑ | LRS↑ |
| SRe2L | 56.3 | -1.5 | 16.2 | 55.0 | -15.6 | 14.2 | 53.4 | -13.2 | 14.8 |
| RDED | **55.7** | **1.6** | **16.8** | **50.2** | -0.6 | **17.4** | **39.8** | -3.6 | **22.9** |
| D4M | 55.9 | -0.6 | 15.6 | 53.0 | -7.7 | 15.8 | 43.7 | -5.8 | 17.6 |
| DWA | 56.1 | -1.2 | 16.3 | 54.4 | -4.1 | 16.1 | 49.7 | -7.8 | 16.3 |
| CDA | 56.2 | -2.5 | 16.1 | 54.9 | -8.6 | 15.3 | 52.0 | -6.7 | 16.1 |
| EDC | 55.7 | -0.8 | 16.4 | 52.0 | **-0.4** | 17.1 | 41.3 | **-0.1** | 18.9 |
| G-VBSM | 56.3 | -1.2 | 16.3 | 55.0 | -7.3 | 15.5 | 44.9 | -5.9 | 17.4 |

Table 6: LRS, HLR, and IOR evaluation results on ImageNet1K. Notably, existing DD methods (mainly decoupled) hardly outperform random selection and perform, and fail to perform well when switched to hard labels.

| IPC | 1 | | | 10 | | | 50 | | |
|---|---|---|---|---|---|---|---|---|---|
| Metric | IOR w/o aug↑ | IOR w/ aug ↑ | ARS↑ | IOR w/o aug↑ | IOR w/ aug ↑ | ARS↑ | IOR w/o aug↑ | IOR w/ aug ↑ | ARS↑ |
| SRe2L | -1.2 | -1.5 | 26.3 | -4.4 | -15.6 | 22.9 | -21.0 | -13.2 | 20.2 |
| RDED | 0.8 | **1.6** | 27.4 | 5.6 | -0.6 | 28.1 | 2.0 | -3.6 | 26.7 |
| D4M | -0.3 | -0.6 | 26.7 | -0.5 | -7.7 | 25.2 | -2.0 | -5.8 | 25.3 |
| DWA | -1.2 | -1.2 | 26.4 | -4.0 | -4.1 | 25.2 | -13.0 | -7.8 | 22.7 |
| CDA | -1.1 | -2.5 | 26.1 | -4.9 | -8.6 | 24.1 | -14.1 | -6.7 | 22.7 |
| EDC | -0.5 | -0.8 | 26.6 | -0.3 | **-0.4** | 26.8 | -3.2 | **-0.1** | 26.2 |
| G-VBSM | -1.2 | -1.2 | 26.4 | -7.9 | -7.3 | 23.8 | -18.0 | -5.9 | 22.1 |

Table 7: Augmentation-robust score (ARS) evaluation results on ImageNet1K. We report both IOR w/ aug ($acc_{syn-aug} - acc_{rdm-aug}$) and IOR w/o aug ($acc_{syn-naug} - acc_{rdm-naug}$). $\gamma$ is 0.5 by default.

data quality, as evidenced by superior LRS scores. This demonstrates that integrating soft labels into the training process can meaningfully enhance synthetic data quality.

*iii) Matching-based methods remain the strongest baselines.* Despite computational limitations that restrict their scalability to large-scale datasets like ImageNet1K, matching-based methods (encompassing gradient, trajectory, and feature matching) consistently produce more effective distilled datasets. Besides, RDED and D4M appear to be more effective among decoupled methods, implying the importance of the realism of synthetic data.

## 4.3 AUGMENTATION-ROBUST SCORE

Table 7 presents ARS performance metrics for various DD methods applied to ImageNet1K, including IOR results with and without data augmentation as introduced in Section 3.3. Most existing decoupled and generative DD methods fail to surpass random selection regardless of augmentation status. Without data augmentation, the performance disparity between DD methods and random selection widens as IPC increases. These findings demonstrate that contemporary DD approaches, despite their heavy reliance on data augmentation strategies, frequently underperform when these same augmentation techniques are applied to simple random selection. Notably, when augmentation is excluded from evaluation, the performance gap between certain DD methodologies and random selection becomes more pronounced, further supporting our assertion that conventional test accuracy metrics no longer serve as an equitable evaluation criterion in this domain.

## 4.4 ANALYSIS

**Robust to model architecture.** Cross-architecture evaluation is an important experiment for dataset distillation methods. Specifically, different models architectures are used to evaluate the synthetic data. Despite variations in raw test accuracy across model architectures, the metric used to evaluate dataset distillation performance should remain consistent, with minimal fluctuation in metric values. As shown in Figure 2, the LRS results for six methods across different settings, each tested with four distinct model architectures, demonstrate high consistency. This consistency validates the robustness of our benchmark across different model architectures.

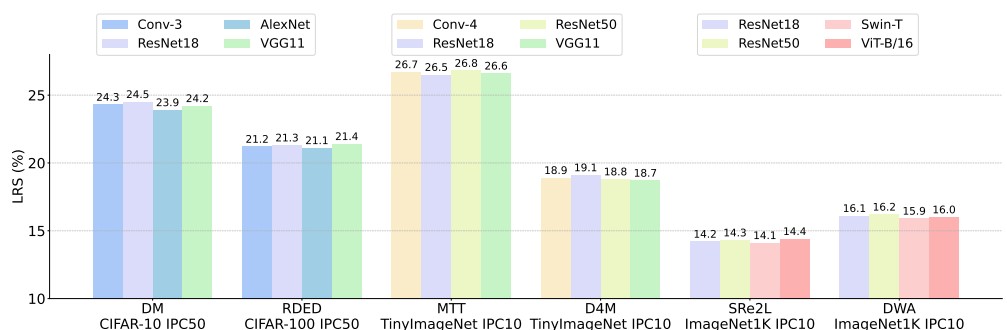

Figure 2: Label-robust scores of methods with four different model architectures. The LRS fluctuation is minimal, indicating that DD-Ranking is robust to different model architectures.

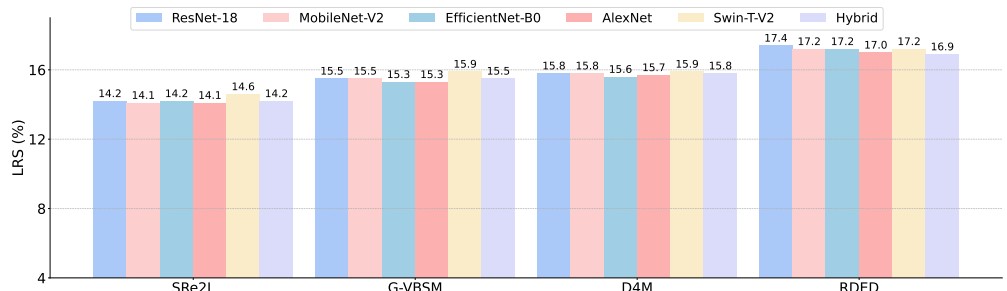

Figure 3: Label-robust scores of decoupled distillation methods with different teacher models. The LRS fluctuation is minimal for each method, indicating that DD-Ranking is robust to soft labels generated by different teahcer models.

**Robust to soft labels.** In decoupled dataset distillation (Shao et al., 2024a; Su et al., 2024b; Sun et al., 2024; Yin et al., 2024), epoch-wise soft labels constitute a crucial component of the synthetic dataset. Recent studies (Cui et al., 2025; Shao et al., 2024a;c) have explored improving test accuracy by leveraging stronger teacher models to provide soft labels without altering the synthetic data itself. However, the validity of this technique remains insufficiently investigated. As shown in Figure 3, whether through the use of different teacher models or advanced hybrid soft label strategies by fusing soft labels generated by multiple teachers, our proposed LRS consistently exhibits strong robustness, thereby validating its reliability across diverse soft label settings.

## 5 RELATED WORKS

**Hard-label-based dataset distillation methods.** Hard-label-based DD methods assign categorical labels to synthetic samples, matching real dataset labels. Matching-based methods are the main hard-label approaches: i) Gradient matching: Optimizes synthetic data to match gradients induced on neural networks. Following DC (Zhao et al., 2021), improvements include DSA (Zhao and Bilen, 2021b), DCC (Lee et al., 2022), and LCMat (Shin et al., 2023). ii) Trajectory matching: Aligns training dynamics between models trained on synthetic and real data. MTT (Cazenavette et al., 2022) introduced this approach, with enhancements from TESLA (Cui et al., 2023), FTD (Du et al., 2023a), and ATT (Liu et al., 2024) improving memory efficiency and reducing trajectory errors. iii) Feature matching: Optimizes synthetic data to produce similar internal representations as real data. Methods like CAFE (Wang et al., 2022a), DM Zhao and Bilen (2023), and DataDAM (Sajedi et al., 2023) provide lightweight frameworks with comparable performance, especially for large IPC settings.

**Soft-label-based dataset distillation methods**. Soft-label DD methods use knowledge distillation during evaluation, with synthetic samples assigned soft labels from pretrained teacher models. Matching-based methods like DATM (Guo et al., 2023), PAD (Li et al., 2024), and EDF (Wang et al.,

2025a) jointly optimize soft labels and synthetic data during trajectory matching. Decoupled methods demonstrate strong scalability on large datasets by decoupling bi-level optimization. SRe2L (Yin et al., 2024) proposed a three-stage "squeeze, recover, and relabel" paradigm, generating and saving soft labels for synthetic samples. Subsequent works CDA (Yin and Shen, 2024), DWA (Sun et al., 2024), EDC (Shao et al., 2024c), and G-VBSM (Shao et al., 2024b) further improve performance from data and soft label perspectives. RDED synthesizes data by concatenating core image patches, while D4M employs diffusion models for high-quality synthetic images.

**Dataset distillation benchmark.** A notable challenge for dataset distillation is the lack of comprehensive benchmarks. DC-Bench (Cui et al., 2022) is the first large-scale standardized benchmark for dataset condensation methods in general. It provides a comprehensive evaluation for several dataset distillation methods and coreset selection methods. Comp-DD is proposed in EDF (Wang et al., 2025a) targeting dataset distillation in complex scenarios. It extracts new subsets from ImageNet1K based on the complexity metric. However, both benchmarks no longer satisfy the need for fair evaluation of dataset distillation methods under the soft label trend. Therefore, we propose DD-Ranking to solve this problem. While distillation for other modalities remains under-explored, we are aware of recent progress on multi-modal dataset distillation (Yim et al., 2025; Zhang et al., 2025; Zhao et al., 2025b). We will track the latest progress in dataset distillation across all areas and enrich our benchmark to be more comprehensive.

## 6 CONCLUSION AND FUTURE WORK

We propose DD-Ranking, a new benchmark that provides a fair and comprehensive evaluation for dataset distillation. DD-Ranking is well motivated by the unfairness originated from inconsistent training settings of existing DD evaluation, especially the use of soft labels and data augmentation. To this end, DD-Ranking introduces both label robust score and augmentation robust score to disentangle the effect of knowledge distillation via soft labeling and data augmentation, and ultimately reveal the true informativeness of distilled datasets. Hopefully, DD-Ranking can facilitate the development of dataset distillation towards improving data quality instead of accuracy. One potential limitation of the current DD-Ranking is that we only support methods for image classification dataset distillation. We are aware that several works (Wu et al., 2024; Zhou et al., 2023) have extended dataset distillation to other tasks and modalities. In the future, we will constantly integrate more baseline methods into our benchmark and extend DD-Ranking to other modalities.

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

# A ADDITIONAL EXPERIMENT RESULTS

Results from Table 3 to Table 6 are computed by letting $\lambda = 0.5$. By default, we treat the hard label recovery and improvement over random equally important. In Table 8 to Table 19, we report the LRS results under different $\lambda$. A larger $\lambda$ gives higher priority to IOR, and a smaller $\lambda$ focuses more on HLR. We encourage future newly proposed DD methods to enhance both HLR and IOR. From Table 20c to Table 20d, we provide the standard deviations for all benchmark results computed under 5 runs with different random seeds. As shown in the table, our ranking is highly consistent, further demonstrating the robustness of our proposed metrics to fairly compare different methods. Also, the LRS score with different $\lambda$ can reflect different methods advantage. For example (Table 18), when $\lambda$ is large (large weight on IOR), RDED's hard-label advantage is diminished (EDC now ranks first jointly with RDED). This aligns with our hard-label evaluation results and the visual quality (RDED selects real image patches and combines them to form one synthetic image, thus performs better under hard labels).

| $\lambda$ | 0.1 | 0.3 | 0.5 | 0.7 | 0.9 |
|---|---|---|---|---|---|
| DC | 11.2 | 14.9 | 19.1 | 24.0 | 29.5 |
| DSA | 9.7 | 13.7 | 18.2 | 23.5 | 29.5 |
| MTT | 14.3 | 18.7 | 23.9 | 29.8 | 36.6 |
| DM | 9.0 | 12.8 | 17.0 | 22.0 | 27.6 |
| DATADAM | 11.9 | 15.8 | 20.2 | 25.2 | 30.9 |
| DATM | 14.4 | 19.2 | 24.6 | 30.9 | 38.2 |
| SRe2L | 7.0 | 10.4 | 14.3 | 18.8 | 23.9 |
| RDED | 9.1 | 12.4 | 16.2 | 20.4 | 25.3 |
| D4M | 11.4 | 14.7 | 18.4 | 22.6 | 27.3 |

Table 8: LRS evaluation results on CIFAR-10 IPC1 under different $\lambda$.

| $\lambda$ | 0.1 | 0.3 | 0.5 | 0.7 | 0.9 |
|---|---|---|---|---|---|
| DC | 15.5 | 19.1 | 23.2 | 27.7 | 32.8 |
| DSA | 16.0 | 19.6 | 23.7 | 28.3 | 33.4 |
| MTT | 19.8 | 23.9 | 28.5 | 33.5 | 39.2 |
| DM | 14.7 | 18.2 | 22.2 | 26.7 | 31.6 |
| DATADAM | 16.1 | 19.7 | 23.8 | 28.4 | 33.5 |
| DATM | 19.0 | 23.5 | 28.7 | 34.5 | 41.2 |
| SRe2L | 7.3 | 10.4 | 13.8 | 17.7 | 22.1 |
| RDED | 11.3 | 14.3 | 17.5 | 21.2 | 25.2 |
| D4M | 14.3 | 17.4 | 20.8 | 24.6 | 28.7 |

Table 9: LRS evaluation results on CIFAR-10 IPC10 under different $\lambda$.

# B ADDITIONAL RELATED WORK

We acknowledge that DD-Ranking has not included enough dataset distillation methods. We discuss them here. In the near future, we will continue to extend our benchmark and include more baseline methods.

# C ETHICS STATEMENT

This work introduces DD-Ranking, a new benchmark for evaluating dataset distillation methods. Our research adheres to the ICLR Code of Ethics and raises minimal ethical concerns. We exclusively use widely-adopted, publicly available benchmark datasets (CIFAR-10, CIFAR-100, TinyImageNet,

| $\lambda$ | 0.1 | 0.3 | 0.5 | 0.7 | 0.9 |
|---|---|---|---|---|---|
| DC | 18.3 | 21.1 | 24.0 | 27.2 | 30.6 |
| DSA | 18.0 | 20.6 | 23.5 | 26.7 | 30.1 |
| MTT | 21.8 | 24.7 | 27.8 | 31.1 | 34.7 |
| DM | 18.7 | 21.4 | 24.3 | 27.5 | 30.9 |
| DATADAM | 19.8 | 22.6 | 25.6 | 28.8 | 32.3 |
| DATM | 21.1 | 24.4 | 28.0 | 31.9 | 36.1 |
| SRe2L | 8.3 | 11.2 | 14.4 | 18.0 | 22.0 |
| RDED | 15.1 | 17.3 | 19.6 | 22.1 | 24.8 |
| D4M | 17.9 | 20.3 | 22.8 | 25.4 | 28.3 |

Table 10: LRS evaluation results on CIFAR-10 IPC50 under different $\lambda$.

| $\lambda$ | 0.1 | 0.3 | 0.5 | 0.7 | 0.9 |
|---|---|---|---|---|---|
| DC | 14.4 | 17.5 | 20.8 | 24.4 | 28.5 |
| DSA | 12.7 | 16.0 | 19.6 | 23.7 | 28.2 |
| MTT | 15.9 | 19.3 | 23.1 | 27.4 | 32.1 |
| DM | 12.1 | 15.3 | 18.9 | 22.8 | 27.2 |
| DATADAM | 12.9 | 16.2 | 19.9 | 23.9 | 28.5 |
| DATM | 19.2 | 22.3 | 25.7 | 29.4 | 33.4 |
| SRe2L | 10.8 | 13.6 | 16.7 | 20.2 | 24.0 |
| RDED | 12.6 | 15.2 | 18.1 | 21.3 | 24.8 |
| D4M | 16.9 | 19.7 | 22.7 | 25.9 | 29.5 |

Table 11: LRS evaluation results on CIFAR-100 IPC1 under different $\lambda$.

and ImageNet) that have been extensively vetted by the research community. No new datasets were collected, and no human subjects were involved in this research.

## D    REPRODUCIBILITY STATEMENT

We have taken extensive measures to ensure the reproducibility of our benchmark. The complete codebase for DD-Ranking, including all evaluation scripts, configuration files, and detailed documentation, will be made publicly available. All experiments use publicly available datasets. All baseline methods we evaluated are publicly available. We have replicated their code and conducted experiments multiple times to verify their reported results.

## E    USE OF LLM

In this work, Large language models were used solely for grammar checking and minor language polishing during the preparation of this manuscript.

| $\lambda$ | 0.1 | 0.3 | 0.5 | 0.7 | 0.9 |
|---|---|---|---|---|---|
| DC | 18.6 | 21.3 | 24.3 | 27.4 | 30.8 |
| DSA | 18.4 | 21.3 | 24.3 | 27.6 | 31.2 |
| MTT | 21.3 | 24.3 | 27.5 | 30.9 | 34.7 |
| DM | 17.1 | 19.9 | 23.0 | 26.2 | 29.8 |
| DATADAM | 18.6 | 21.5 | 24.6 | 28.0 | 31.7 |
| DATM | 20.9 | 23.7 | 26.8 | 30.1 | 33.6 |
| SRe2L | 11.0 | 12.9 | 15.0 | 17.3 | 19.8 |
| RDED | 14.7 | 17.0 | 19.4 | 22.0 | 24.9 |
| D4M | 14.3 | 17.4 | 20.9 | 24.7 | 29.0 |

Table 12: LRS evaluation results on CIFAR-100 IPC10 under different $\lambda$.

| $\lambda$ | 0.1 | 0.3 | 0.5 | 0.7 | 0.9 |
|---|---|---|---|---|---|
| DC | 19.4 | 21.0 | 22.7 | 24.5 | 26.4 |
| DSA | 19.6 | 21.2 | 23.0 | 24.8 | 26.8 |
| MTT | 22.9 | 24.8 | 26.8 | 28.8 | 31.0 |
| DM | 21.3 | 23.1 | 24.9 | 26.9 | 29.0 |
| DATADAM | 22.9 | 24.8 | 26.8 | 28.9 | 31.1 |
| DATM | 24.2 | 27.2 | 30.4 | 33.9 | 37.6 |
| SRe2L | 12.1 | 14.1 | 16.2 | 18.5 | 21.0 |
| RDED | 17.6 | 19.3 | 21.2 | 23.1 | 25.2 |
| D4M | 18.3 | 21.1 | 24.2 | 27.5 | 31.1 |

Table 13: LRS evaluation results on CIFAR-100 IPC50 under different $\lambda$.

| $\lambda$ | 0.1 | 0.3 | 0.5 | 0.7 | 0.9 |
|---|---|---|---|---|---|
| DC | 17.4 | 19.6 | 22.0 | 24.5 | 27.2 |
| DSA | 16.9 | 19.1 | 21.6 | 24.2 | 27.0 |
| MTT | 16.8 | 19.3 | 21.9 | 24.8 | 27.8 |
| DM | 15.0 | 17.5 | 20.2 | 23.1 | 26.2 |
| DATM | 18.5 | 20.9 | 23.5 | 26.2 | 29.2 |
| EDF | 18.4 | 20.9 | 23.5 | 26.3 | 29.4 |
| SRe2L | 12.5 | 15.1 | 17.9 | 21.0 | 24.3 |
| RDED | 15.8 | 18.3 | 20.9 | 23.8 | 26.9 |
| D4M | 13.8 | 16.1 | 18.6 | 21.2 | 24.1 |

Table 14: LRS evaluation results on TinyImageNet IPC1 under different $\lambda$.

| $\lambda$ | 0.1 | 0.3 | 0.5 | 0.7 | 0.9 |
|---|---|---|---|---|---|
| DC | 19.7 | 21.7 | 23.9 | 26.3 | 28.7 |
| DSA | 20.0 | 22.0 | 24.1 | 26.3 | 28.7 |
| MTT | 21.9 | 24.2 | 26.7 | 29.3 | 32.1 |
| DM | 18.2 | 20.6 | 23.1 | 25.8 | 28.7 |
| DATM | 20.9 | 23.4 | 26.0 | 28.8 | 31.8 |
| EDF | 20.9 | 23.5 | 26.2 | 29.2 | 32.3 |
| SRe2L | 12.8 | 14.9 | 17.1 | 19.5 | 22.1 |
| RDED | 18.2 | 20.1 | 22.1 | 24.2 | 26.5 |
| D4M | 15.1 | 16.9 | 18.9 | 21.1 | 23.3 |

Table 15: LRS evaluation results on TinyImageNet IPC10 under different $\lambda$.

| $\lambda$ | 0.1 | 0.3 | 0.5 | 0.7 | 0.9 |
|---|---|---|---|---|---|
| DC | 19.4 | 20.8 | 22.2 | 23.7 | 25.2 |
| DSA | 20.9 | 22.8 | 24.8 | 26.9 | 29.1 |
| MTT | 21.7 | 23.7 | 25.8 | 28.0 | 30.3 |
| DM | 20.4 | 22.2 | 24.1 | 26.1 | 28.1 |
| DATM | 22.6 | 24.9 | 27.2 | 29.8 | 32.4 |
| EDF | 22.5 | 24.9 | 27.3 | 30.0 | 32.8 |
| SRe2L | 15.5 | 17.0 | 18.6 | 20.3 | 22.1 |
| RDED | 21.4 | 22.5 | 23.7 | 24.8 | 26.0 |
| D4M | 17.9 | 20.8 | 23.8 | 27.2 | 30.8 |

Table 16: LRS evaluation results on TinyImageNet IPC50 under different $\lambda$.

| $\lambda$ | 0.1 | 0.3 | 0.5 | 0.7 | 0.9 |
|---|---|---|---|---|---|
| SRe2L | 9.9 | 12.9 | 16.2 | 19.9 | 24.0 |
| RDED | 10.2 | 13.3 | 16.8 | 20.8 | 25.2 |
| D4M | 10.1 | 13.1 | 16.4 | 20.2 | 24.4 |
| DWA | 10.0 | 13.0 | 16.3 | 20.0 | 24.1 |
| CDA | 9.9 | 12.8 | 16.1 | 19.7 | 23.7 |
| EDC | 10.1 | 13.1 | 16.4 | 20.1 | 24.3 |
| G-VBSM | 10.0 | 12.9 | 16.3 | 20.0 | 24.1 |

Table 17: LRS evaluation results on ImageNet1K IPC1 under different $\lambda$.

| $\lambda$ | 0.1 | 0.3 | 0.5 | 0.7 | 0.9 |
|---|---|---|---|---|---|
| SRe2L | 9.9 | 12.0 | 14.2 | 16.7 | 19.3 |
| RDED | 11.4 | 14.2 | 17.4 | 20.8 | 24.6 |
| D4M | 10.6 | 13.0 | 15.8 | 18.7 | 22.0 |
| DWA | 10.3 | 13.1 | 16.1 | 19.5 | 23.2 |
| CDA | 10.1 | 12.6 | 15.3 | 18.3 | 21.6 |
| EDC | 11.0 | 13.9 | 17.1 | 20.6 | 24.6 |
| G-VBSM | 10.1 | 12.7 | 15.5 | 18.6 | 22.1 |

Table 18: LRS evaluation results on ImageNet1K IPC10 under different $\lambda$.

| $\lambda$ | 0.1 | 0.3 | 0.5 | 0.7 | 0.9 |
|---|---|---|---|---|---|
| SRe2L | 10.3 | 12.5 | 14.8 | 17.4 | 20.2 |
| RDED | 14.0 | 16.2 | 18.6 | 21.2 | 23.9 |
| D4M | 12.9 | 15.1 | 17.6 | 20.2 | 23.0 |
| DWA | 11.3 | 13.7 | 16.3 | 19.1 | 22.1 |
| CDA | 10.8 | 13.3 | 16.1 | 19.1 | 22.4 |
| EDC | 13.7 | 16.2 | 18.9 | 21.9 | 25.1 |
| G-VBSM | 12.6 | 14.9 | 17.4 | 20.0 | 22.9 |

Table 19: LRS evaluation results on ImageNet1K IPC50 under different $\lambda$.

| ipc | 1 | | | 10 | | | 50 | | |
|---|---|---|---|---|---|---|---|---|---|
| metric | HLR↓ | IOR↑ | LRS↑ | HLR↓ | IOR↑ | LRS↑ | HLR↓ | IOR↑ | LRS↑ |
| DC | 0.2 | 0.2 | 0.3 | 0.3 | 0.2 | 0.3 | 0.2 | 0.4 | 0.3 |
| DSA | 0.3 | 0.3 | 0.2 | 0.4 | 0.2 | 0.2 | 0.5 | 0.4 | 0.4 |
| MTT | 0.5 | 0.7 | 0.6 | 0.6 | 0.8 | 0.8 | 0.5 | 0.4 | 0.5 |
| DM | 0.7 | 0.6 | 0.5 | 0.8 | 0.9 | 1.0 | 0.7 | 0.7 | 0.7 |
| DATADAM | 0.8 | 0.5 | 0.6 | 0.6 | 0.6 | 0.5 | 0.7 | 0.5 | 0.6 |
| DATM | 0.7 | 0.4 | 0.8 | 0.5 | 0.7 | 0.6 | 0.3 | 0.7 | 0.5 |
| SRe2L | 0.5 | 0.6 | 0.6 | 0.8 | 0.8 | 0.6 | 0.5 | 0.8 | 0.7 |
| RDED | 0.7 | 0.9 | 0.7 | 0.8 | 0.7 | 0.8 | 0.9 | 1.2 | 0.9 |
| D4M | 0.8 | 0.8 | 0.6 | 0.7 | 0.9 | 0.9 | 0.8 | 1.0 | 0.9 |

(a) Standard deviations of LRS results on CIFAR-10.

| ipc | 1 | | | 10 | | | 50 | | |
|---|---|---|---|---|---|---|---|---|---|
| metric | HLR↓ | IOR↑ | LRS↑ | HLR↓ | IOR↑ | LRS↑ | HLR↓ | IOR↑ | LRS↑ |
| DC | 0.4 | 0.3 | 0.3 | 0.3 | 0.5 | 0.4 | 0.2 | 0.6 | 0.7 |
| DSA | 0.5 | 0.5 | 0.5 | 0.4 | 0.4 | 0.3 | 0.4 | 0.5 | 0.4 |
| MTT | 0.5 | 0.6 | 0.5 | 0.7 | 0.7 | 0.6 | 0.5 | 0.6 | 0.6 |
| DM | 0.6 | 0.8 | 0.7 | 0.9 | 0.9 | 0.9 | 0.7 | 0.7 | 0.5 |
| DATADAM | 0.6 | 0.7 | 0.8 | 0.5 | 0.8 | 0.7 | 0.7 | 0.6 | 0.6 |
| DATM | 0.7 | 0.5 | 0.6 | 0.6 | 0.6 | 0.6 | 0.5 | 0.8 | 0.7 |
| SRe2L | 1.1 | 0.9 | 0.9 | 0.8 | 0.7 | 0.7 | 0.5 | 0.9 | 0.7 |
| RDED | 0.7 | 1.0 | 0.8 | 0.6 | 0.9 | 0.6 | 0.8 | 1.1 | 0.9 |
| D4M | 0.5 | 0.6 | 0.4 | 0.7 | 0.8 | 0.6 | 0.8 | 0.9 | 0.9 |

(b) Standard deviations of LRS results on CIFAR-100.

| ipc | 1 | | | 10 | | | 50 | | |
|---|---|---|---|---|---|---|---|---|---|
| metric | HLR↓ | IOR↑ | LRS↑ | HLR↓ | IOR↑ | LRS↑ | HLR↓ | IOR↑ | LRS↑ |
| DC | 0.4 | 0.3 | 0.3 | 0.2 | 0.5 | 0.5 | 0.4 | 0.6 | 0.4 |
| DSA | 0.5 | 0.4 | 0.6 | 0.3 | 0.7 | 0.4 | 0.5 | 0.5 | 0.4 |
| MTT | 0.5 | 0.6 | 0.6 | 0.4 | 0.7 | 0.6 | 0.3 | 0.8 | 0.6 |
| DM | 0.3 | 0.7 | 0.5 | 0.8 | 0.9 | 0.8 | 0.5 | 0.8 | 0.7 |
| DATM | 0.5 | 0.4 | 0.4 | 0.3 | 0.5 | 0.6 | 0.4 | 0.9 | 0.6 |
| SRe2L | 0.7 | 0.7 | 0.7 | 0.5 | 0.4 | 0.5 | 0.4 | 0.8 | 0.8 |
| RDED | 0.6 | 0.8 | 0.7 | 0.5 | 0.9 | 0.7 | 0.7 | 1.0 | 0.9 |
| D4M | 0.6 | 0.7 | 0.6 | 0.7 | 0.8 | 0.6 | 0.9 | 1.1 | 0.8 |

(c) Standard deviations of LRS results on TinyImageNet.

| ipc | 1 | | | 10 | | | 50 | | |
|---|---|---|---|---|---|---|---|---|---|
| metric | HLR↓ | IOR↑ | LRS↑ | HLR↓ | IOR↑ | LRS↑ | HLR↓ | IOR↑ | LRS↑ |
| SRe2L | 0.6 | 1.1 | 0.8 | 0.5 | 0.8 | 0.9 | 0.7 | 1.0 | 0.7 |
| RDED | 0.5 | 0.7 | 0.6 | 0.4 | 0.9 | 0.8 | 0.7 | 0.7 | 0.7 |
| D4M | 0.8 | 0.6 | 0.6 | 0.5 | 1.1 | 0.9 | 0.6 | 0.8 | 0.5 |
| DWA | 0.8 | 1.2 | 1.1 | 0.9 | 1.1 | 1.0 | 0.7 | 0.8 | 0.9 |
| CDA | 0.6 | 0.6 | 0.8 | 0.7 | 0.8 | 0.7 | 0.4 | 0.8 | 0.6 |
| EDC | 0.3 | 0.8 | 0.4 | 0.5 | 0.4 | 0.5 | 0.5 | 1.1 | 0.9 |
| G-VBSM | 0.6 | 1.2 | 1.0 | 0.5 | 1.3 | 0.9 | 0.6 | 0.9 | 0.8 |

(d) Standard deviations of LRS results on ImageNet-1K.

| Category | Method |
|---|---|
| Kernel-based | KIP-FC (Nguyen et al., 2021a) |
| | KIP-ConvNet (Nguyen et al., 2021b) |
| | FRePo (Zhou et al., 2022) |
| | RFAD (Loo et al., 2022a) |
| | RCIG (Loo et al., 2023) |
| Gradient-matching | DC (Zhao et al., 2021) |
| | DSA (Zhao and Bilen, 2021b) |
| | DCC (Lee et al., 2022) |
| | LCMat (Shin et al., 2023) |
| Trajectory-matching | MTT (Cazenavette et al., 2022) |
| | TESLA (Cui et al., 2023) |
| | FTD (Du et al., 2023a) |
| | SeqMatch (Du et al., 2023b) |
| | DATM (Guo et al., 2022) |
| | ATT (Liu et al., 2024) |
| | NSD (Yang et al., 2024) |
| | PAD (Li et al., 2024) |
| | EDF (Wang et al., 2025a) |
| | SelMatch (Lee and Chung, 2024) |
| Distribution-matching | DM (Zhao and Bilen, 2021a) |
| | CAFE (Wang et al., 2022b) |
| | IDM (Zhao et al., 2023) |
| | DREAM (Liu et al., 2023) |
| | M3D (Zhang et al., 2023) |
| | NCFD (Wang et al., 2025b) |
| Generative model | DiM Wang et al. (2023) |
| | GLaD (Cazenavette et al., 2023) |
| | H-PD (Zhong et al., 2024) |
| | LD3M (Moser et al., 2024) |
| | IT-GAN (Zhao and Bilen, 2022b) |
| | D4M Su et al. (2024a) |
| | Minimax Diffusion Gu et al. (2023) |
| Decoupled | SRe2L (Yin et al., 2024) |
| | RDED (Sun et al., 2024) |
| | HeLIO (Yu et al., 2024) |
| | DWA (Du et al., 2024b) |
| | CDA (Yin and Shen, 2023) |
| | EDC (Shao et al., 2024c) |
| | G-VBSM (Shao et al., 2024a) |
| Others | MIM4DD (Shang et al., 2023) |
| | DQAS (Zhao et al., 2024) |
| | LDD (Zhao et al., 2025a) |

Table 21: Summary of previous works on dataset distillation

