# OpenReview forum: "DD-Ranking: Rethinking the Evaluation of Dataset Distillation"
_ICLR.cc/2026/Conference — Submitted to ICLR 2026_

### Official Review · Reviewer_6gJb · 2025-10-23

**Soundness:** 2
**Presentation:** 2
**Contribution:** 2
**Rating:** 4
**Confidence:** 5

**Summary:**

This paper first observes that existing dataset distillation approaches adopt inconsistent evaluation protocols, differing in their use of label types (e.g., fixed hard/soft labels or per-image soft labels), data augmentation strategies, etc. To enable a fair comparison across methods, the authors propose a unified benchmark along with two new evaluation metrics, LRS and ARS, designed to be robust against variations in the use of label types and augmentation techniques. Experimental results reveal that recent approaches relying heavily on soft labels (i.e., decoupled methods) are ineffective, often performing worse than a random selection baseline. In contrast, more conventional methods employing a single hard or soft label per image continue to demonstrate superior performance.

**Strengths:**

- The introduction of a unified benchmark for dataset distillation methods is both timely and significant for the community. Establishing a standardized evaluation protocol is crucial for ensuring fair comparisons and clear assessments across approaches. The paper convincingly shows that recent methods relying on per-epoch soft labels have been overrated.
- The paper proposes novel evaluation metrics, LRS and ARS. These metrics provide a more robust and flexible framework for evaluating methods that adopt different recipes to train networks on distilled datasets.
- The paper provides extensive experimental results.

**Weaknesses:**

- The main finding that improvements in decoupled methods come largely from knowledge distillation rather than the distilled dataset itself is interesting, but this point has already been raised in [1]. It would be helpful to acknowledge and connect to that prior work.
- Some experimental results appear to be missing. For example, lines 149–161 mention results on random noise with soft labels, but I could not find the corresponding figure.
- While comparing against randomly selected samples is meaningful, I am less convinced by the comparisons using hard labels or without augmentation. Many methods are explicitly designed with particular label formats and augmentation strategies (e.g., DATM, EDF, IDC, FYI, etc.), so the evaluation feels less fair in those settings.
- The evaluation metrics introduce tunable parameters ($\lambda$ in Eq. (3) and $\gamma$ in Eq. (4)). This flexibility may unintentionally influence the results and deserves some discussion.
- A few methods evaluated in Sec. 4.2 are not included in Sec. 4.3, which leaves the comparison incomplete.

[1]: Qin, Tian, Zhiwei Deng, and David Alvarez-Melis. "A label is worth a thousand images in dataset distillation." Advances in Neural Information Processing Systems 37 (2024): 131946-131971.

**Questions:**

- How were the soft labels generated for random images when comparing with DATM or EDF? In the original methods, labels are jointly optimized with the images. Clarification on this process would be helpful.
- Why are learning rates tuned specifically for randomly selected images (lines 267–269) instead of applying the same learning rate used with synthetic images? Please explain the rationale behind this choice.

---

> ### Author Response · Authors · 2025-11-19
> **Rebuttal First Page**
>
> **W1:** The main finding that improvements in decoupled methods come largely from knowledge distillation rather than the distilled dataset itself is interesting, but this point has already been raised in [1]. It would be helpful to acknowledge and connect to that prior work.
>
> **R1:** Thank you for your suggestion. We acknowledge that previous works have pointed out that label-based optimization can lead to better performance when performing large-scale dataset distillation. **We have added the acknowledgement at line 163-164 of the revision**.
>
> Our work highlights that existing dataset distillation methods often use inconsistent post-evaluation setups during comparisons, including inconsistent soft label setting, leading to impure improvements in test accuracy.  Therefore, we introduce a novel evaluation benchmark and demonstrate the true effects of existing methods. Additional experimental results, as shown in Figures 2 and Figures 3, further validate the generalization and robustness of our proposed metric, providing a new and reliable evaluation framework for the performance assessment of future work.
>
> ---
>
>
> **W2:** Some experimental results appear to be missing. For example, lines 149–161 mention results on random noise with soft labels, but I could not find the corresponding figure.
>
> **R2:** We are very sorry for the missing results. **We have updated in the revision Table 2a** and also paste it here.
>
> | Dataset     | Random Guessing | Batchsize=128 | Batchsize=1024 |
> | ----------- | --------------- | ------------- | -------------- |
> | ImageNet-1K | 0.1%            | 0.5%          | 0.3%           |
>
> Under different batch sizes, the pure noise outperforms random guessing. These results provide strong evidence for our claim that soft labels (used at the evaluation time of DD) don't contribute to data quality, but only improve the test accuracy.
>
> Additionally, we provide the performance of a model trained on noisy images with soft labels as follows (the figures in the brackets represent the improvement over hard labels). The training setting is kept the same as above.
>
> | dataset      | noise std = 0.05 | noise std = 0.5 |
> | :----------- | :--------------- | :-------------- |
> | TinyImageNet | 49.9% ( 26.0%)   | 14.6% ( 3.9%)   |
> | ImageNet1K   | 60.6% ( 24.9%)   | 6.8% ( 5.6%)    |
>
> As shown in the table, under both light (std=0.1) and heavy noise (std=0.5), the model trained on these noisy images with soft labels could still achieve remarkable performance, and significantly outperform the one trained with hard labels. These results further demonstrate the effectiveness of soft labels in enhancing model training and improving the test accuracy.
>
> ---
>
> **W3:** While comparing against randomly selected samples is meaningful, I am less convinced by the comparisons using hard labels or without augmentation. Many methods are explicitly designed with particular label formats and augmentation strategies (e.g., DATM, EDF, IDC, FYI, etc.), so the evaluation feels less fair in those settings.
>
> **R3:** Thanks for the question. We use hard label accuracy as part of our evaluation metric because training models on the original dataset does not require specific labels. On one hand, to claim how much the synthetic dataset **recovers** the original dataset information, it should be evaluated under the same setting. Otherwise, it's still **not fair** especially when some methods claim "achieving lossless performance" since knowledge distillation or data augmentation can also improve the performance on the original dataset. On the other hand, we believe a distilled dataset is informative **if and only if it has informative samples** (image in this case) that can bring decent hard-label performance. If the performance relies heavily on the label design or augmentation design, then there are two major concerns: 1) it's some data-irrelevant factors that improves the performance, such as teacher knowledge 2) the distilled dataset is not robust and cannot be practically used in other downstream tasks, such as object detection or segmentation.
>
> We acknowledge that previous work has optimized labels to achieve further performance improvements. However, existing works that deploy soft labels often use different post-evaluation setups for soft labels. Thus, using hard label performance as part of the metric ensures that the final metric is more robust and comprehensive. Additionally, when calculating the performance difference under the soft label setup, we use the soft labels generated by the corresponding method, thereby **accounting for the contribution** of existing methods to soft label performance.
>
> We have also updated a theoretical analysis of DD-Ranking's fairness in the global response. We welcome any further discussion.

---

> ### Author Response · Authors · 2025-11-19
> **Rebuttal Continued**
>
> **W4:** The evaluation metrics introduce tunable parameters ( in Eq. (3) and in Eq. (4)). This flexibility may unintentionally influence the results and deserves some discussion.
>
> **R4:** Thanks for the question. In Equation 3, we use $\lambda$ to adjust the relative weight between HLR and IOR. An empirical setting of 0.5 means we treat both scores equally important. In our public leaderboard to be released, we will use the default setting to demonstrate the ranking. However, we  allow researchers to have flexible evaluation metrics that are more closely aligned with the evaluation needs of different distillation dataset application scenarios. For example, a method that focuses only on hard-label-based dataset distillation is welcomed.
>
> However, we appreciate your suggestion to discuss the impact of of different $\lambda$ or $\gamma$ values. In Appendix A, we have shown the evaluation results under various $\lambda$ and $\gamma$ values. Here, we use an example for discussion: the LRS ranking under different $\lambda$ on ImageNet-1K IPC10.
>
> | $\lambda$ = 0.1 | $\lambda$ = 0.3 | $\lambda$ = 0.5 | $\lambda$ = 0.7 | $\lambda$ = 0.9 |
> | --------------- | --------------- | --------------- | --------------- | --------------- |
> | RDED            | RDED            | RDED            | RDED            | RDED,EDC        |
> | EDC             | EDC             | EDC             | EDC             | DWA             |
> | D4M             | DWA             | DWA             | DWA             | G-VBSM          |
> | DWA             | D4M             | D4M             | D4M             | D4M             |
> | CDA,G-VBSM      | G-VBSM          | G-VBSM          | G-VBSM          | CDA             |
> | SRe2L           | CDA             | CDA             | CDA             | SRe2L           |
> |                 | SRe2L           | SRe2L           | SRe2L           |                 |
>
> As shown in the table, our ranking is **highly consistent**, further demonstrating the robustness of our proposed metrics to fairly compare different methods. Also, the LRS score with different $\lambda$ can reflect different methods advantage. For example, when $\lambda$ is large (large weight on IOR), RDED's hard-label advantage is diminished (EDC now ranks first jointly with RDED). This aligns with our hard-label evaluation results and the visual quality (RDED selects real image patches and combines them to form one synthetic image, thus performs better under hard labels).
>
> **The discussion above has been added to the Appendix of the revision.**
>
> ---
>
> **Q1:** How were the soft labels generated for random images when comparing with DATM or EDF? In the original methods, labels are jointly optimized with the images. Clarification on this process would be helpful.
>
> **A1:** Thanks for the question. When computing the IOR for methods that simultaneously optimize both labels and images, such as EDF and DATM, we generate a **fixed** set of soft labels for random samples using their teacher models. We contend that the joint optimization of distilled images and labels in these approaches is **inherently aligned with the nature of dataset distillation, and can improve the synthetic data**. Moreover, the resulting labels are fixed and remain unchanged during post-evaluation phase. This stands in contrast to decoupled dataset distillation methods, where soft labels of both distilled images and random sampled images are dynamically obtained at each epoch via a teacher model under the knowledge distillation framework.
>
> ---
>
> **Q2:** Why are learning rates tuned specifically for randomly selected images (lines 267–269) instead of applying the same learning rate used with synthetic images? Please explain the rationale behind this choice.
>
> **A2:** Thanks for the question. Through empirical investigation, we observed that the learning rate significantly influences the final test accuracy across different datasets. A poor learning rate could result in highly unreasonable performance. Existing dataset distillation studies often adopt learning rates optimized specifically for their proposed post-evaluation settings; thus, it's crucial to do the same for random samples. On the other hand, existing DD methods didn't adequately consider learning rate configurations in standard hard-label settings. To mitigate the impact of learning rate variations, our approach selects optimal learning rates **not only for randomly chosen images but also for distilled datasets**. This methodological consideration is further explained in lines 267–268.

---

> ### Author Response · Authors · 2025-11-24
> **A summary of rebuttal for your convenience. We appreciate your feedback!**
>
> We thank you again for providing valuable review comments. To save your time, we summarize our response to your questions as follows:
>
> 1. **Discussion of related work:** Thanks for your suggestion. We have updated the revision accordingly.
> 2. **Missing results:** We are truly sorry for the missed results. We have updated the results and added more in the revision. Please find our settings and analysis in our previous response or in the revision.
> 3. **Comparisons using hard labels or without augmentation are less convincing:** Thanks for the question. Our main reason is that soft labels, data augmentation, or any other type of data-irrelevant tricks are not born with the original dataset. Therefore, although methods with these tricks have high accuracy, it is crucial to see if the performance boost is brought by these tricks (what we don't want) or the synthetic data (what we want).
>    However, we are not preventing DD methods from using these tricks, as long as they are using them properly to truly improve the informativeness of synthetic data. And that has been accounted by our proposed LRS and ARS (e.g. DATM still outperforms random samples with soft labels to a large extent).
> 4. **Diccussion of $\lambda$ or $ \gamma$ :** Thanks for the great suggestion. Please find our discussion in the previous response or in the revision. The main takeaway is that the ranking under different $\lambda$  or $ \gamma$ values is highly consistent, and can reveal different methods' advantages.
> 5. **How to generate soft labels for methods like DATM or EDF:** Thanks for the question. This track of methods is distilling synthetic soft labels along with data samples. In that sense, there is a one-to-one mapping between data and label (this is different from decoupled methods, which have one-to-many mappings). Therefore, we generate a fixed soft label for each random sample using the same teacher model.
> 6. **Why tune the learning rate:** Thanks for the question. The main reason is that existing methods have already tuned learning rates to a reasonable range based on their data distribution. Thus, it's crucial to do the same for random selection since a poorly set learning rate can result in unreasonably poor performance. Also, for hard label evaluation on soft label-based methods, learning rate tuning is done for both random selection and the DD method.
>
> If you have any further concerns, don't hesitate to reach us. We are looking forward to your reply!

---

> > ### Comment · Reviewer_6gJb · 2025-11-27
> >
> > Thank you for the detailed response and the revisions to the manuscript. While the updates have addressed some of my points, I still have remaining concerns.
> >
> > **1. Theoretical Rigor**
> >
> > The newly added theoretical foundation requires further clarification to ensure rigor:
> > * Definition of $\epsilon$: What is the justification for defining $\epsilon$ as being sampled from a normal distribution?
> > * Missing Definitions: Several notations appear undefined, such as $ACC(\cdot)$ and terms related to $r$. Explicit definitions are necessary for clarity.
> > * Accuracy Formulation: It is not immediately clear why accuracy can be formulated as a simple summation, as shown in Eq. (5).
> > * Assumption on $\beta(\pi)$: **Crucially**, I have reservations about the assumption that $\beta(\pi)$ is consistent. It is difficult to accept that the performance gain yielded by a training protocol remains constant across different datasets. For instance, methods like IDC split images into four parts and resize them to the original resolution. Such an augmentation strategy is unlikely to yield the same performance gains on standard images (e.g., randomly selected ones) due to the distributional shift.
> >
> > **2. Sensitivity Analysis ($\lambda$ and $\gamma$)**
> >
> > The discussion on $\lambda$ and $\gamma$ is currently limited to LRS. To fully demonstrate the robustness of the proposed method, it is essential to provide consistent results for ARS as well, particularly for methods like IDC.
> >
> > **3. Evaluation Settings (Augmentation)**
> >
> > Regarding my previous concern about comparisons without augmentation: Although I have read your response, I remain unconvinced about the necessity and fairness of this setting. As noted previously, some approaches (e.g., IDC) are explicitly designed to work with specific augmentation strategies. Evaluating them without these components detaches the methods from their intended usage and may not provide meaningful insights. I question the value of comparing methods in a setting where they are structurally disadvantaged.
> >
> > **Minor Point**
> > * The caption for (c) is missing in Table 2.
> >
> > Please let me know if I have any misunderstandings regarding the paper.

---

> ### Author Response · Authors · 2025-11-27
> **Response to Reviewer's Questions (1/2)**
>
> ## Q1: The newly added theoretical foundation requires further clarification to ensure rigor
>
> Thanks very much for your advice. Before we dive into your questions, we want to emphasize that the evaluation of DD remains experiment-heavy, meaning the most effective way is to measure the performance. Thus, our theoretical analysis only provides some intuition here about why DD-Ranking is a more fair benchmark than previously used pure accuracy. We will make such claim in the revision and move the theoretical analysis to the Appendix in the final version.
>
> - **Definition of $ \epsilon $ and what is the justification for defining as being sampled from a normal distribution?** Here, we are treating the accuracy as a random variable given that we will run the evaluation multiple times with different random seeds. Thus, there will be some random fluctuations. We draw it from normal distribution because normal distribution is empirically good for a residual term. However, changing it to any other distribution doesn't affect our derivation since the expectation cancels out.
> - **Missing Definitions: Several notations appear undefined, such as and terms related to . Explicit definitions are necessary for clarity.** We are sorry for the unclear definition. 1) $ACC(D, \pi)$ refers to the accuracy of the the model trained on dataset $D$ under evaluation setting $\pi$. 2) $r_A$ or $r_B$ refers to the random selection baseline under the evaluation setting of method $A$ or $B$. We will update the revision accordingly.
> - **Accuracy Formulation: It is not immediately clear why accuracy can be formulated as a simple summation, as shown in Eq. (5).**
>   1. From a statistical perspective, we are treating this like an analysis of variance (ANOVA) or a linear regression model. We treat the intrinsic Data quality ($q$) and the improvement brought by the evaluation setting ($\beta$) as two distinct factors contributing to the final accuracy. $q$ is the signal we want to measure and $\beta$ is a confounding variable. The additive model allows us to mathematically disentangle these factors.
>   2. However, you are right to question the linear relationship and we are making a simplification to demonstrate the proof. Here, you can think of the additive model represents a first-order Taylor Series expansion of this function around our operating point. By assuming linearity (additivity), we effectively capture the **marginal gain** provided by the synthetic data quality, independent of the training recipe.
> - **Assumption of $\beta(\pi)$** We totally understand your concern and explain in detail as follows:
>   1. First of all, we want to argue that the focus of IDC is more towards reparameterization, instead of data augmentation. The real effect of its reparameterization is to increase the IPC while keeping the storage constant. However, you can also say IDC is designed with specific augmentation and it doesn't conflict with our following explanation.
>   2. We want to clarify that in your described scenario, the true difference lies in $q$ instead of $\pi$. We'd like to differentiate the data augmentation used for distillation and evaluation. For works like DSA[1] which applies differentiable data augmentation during distillation and FYI[2] which embeds a flipping counterpart into synthetic images during training (or IDC[3] that uses multi formation in your view), **their synthetic truly benefits from data augmentation. In these cases, it is $q$ that has been improved, not $\beta$.** And our ARS can capture that by showing the method still outperforms the random selection with the same augmentation.
>   3. However, when we talk about data-irrelevant augmentation in this work, we refer to augmentation techniques that are not specific designs for the distillation algorithm and can be applied to any dataset. **In this case, the improvements are from $\beta$ instead of $q$.** And our proposed ARS is to measure whether the use of augmentation improves the data or just the accuracy.
>
>
> [1] Dataset Condensation with Differentiable Siamese Augmentation, in ICML 2021
>
> [2] FYI: Flip Your Images for Dataset Distillation, in ECCV 2024
>
> [3] Dataset Condensation via Efficient Synthetic-Data Parameterization, in ICML 2022

---

> ### Author Response · Authors · 2025-11-27
> **Response to Reviewer's Question (2/2)**
>
> ## Q2: Discussion of $\gamma$
>
> A2: Thanks for the question. As shown in the table below, ARS under various $\gamma$ also show consistent ranking with minor fluctuation. Here, the comparison between RDED and SRe2L further supports our previous response. Both methods are using random cropping and Cutmix, but RDED with concatenated image patches demonstrates much better distillation quality, matching our ranking.
>
> | $\gamma$ = 0.1 | $\gamma$ = 0.3 | $\gamma$ = 0.5 | $\gamma$ = 0.7 | $\gamma$ = 0.9 |
> | -------------- | -------------- | -------------- | -------------- | -------------- |
> | RDED           | RDED           | RDED           | RDED           | RDED           |
> | EDC            | EDC            | EDC            | EDC            | EDC            |
> | DWA            | DWA            | DWA            | D4M            | D4M            |
> | D4M            | D4M            | D4M            | DWA            | DWA            |
> | G-VBSM         | G-VBSM         | CDA            | CDA            | CDA            |
> | CDA            | CDA            | G-VBSM         | G-VBSM         | G-VBSM         |
> | SRe2L          | SRe2L          | SRe2L          | SRe2L          | SRe2L          |
>
>
>
> ## Q3: Concerns about augmentation
>
> A3: Thanks for raising this concern. As we have explained in the answer to Q1, in methods like DSA, IDC, and FYI, it is $q$ that has been improved, not $\beta$. Thus, they still outperform random selection. We further clarify why we evaluate all DD methods without some components such as data augmentation as follows:
>
> 1. We argue that it's unreasonable to build a dataset with restrictions. That is, to use the dataset, one must apply some technique otherwise it won't give a even reasonable result. As the naming suggests, augmentation should be considered as a good-to-have improvement, not something necessary. We can make an analogy to modern LLMs where people usually do prompt engineering. Indeed, there will be some better system prompts that give us better outputs, but other prompts should still make the LLM work instead of outputting gibberish. Therefore, we believe a distilled dataset should still behave like the original one, instead of adding more and more constraints to it. Unfortunately, we found that many methods nowadays are now relying on these techniques to perform well instead of focusing on synthesizing real high-quality data.
> 2. Also, we do this to make sure that distilled datasets are robust to real-world use. For example, image datasets may not only be used for classification, but it can also be used in segmentation or generative tasks. In this case, if the distilled dataset designs specific constraint to make it work only for classification, its practical impact will be significantly harmed.
> 3. In a summary, we think evaluating them without augmentation is not to make them disadvantageous. Instead, DD methods should maintain a good performance without these techniques to demonstrate their robustness. Otherwise, it deviates from the initial goal of dataset distillation[4].
>
> Again, we are not preventing future methods from applying these techniques. Instead, we are advocating proper use that can truly improve the data quality and robustness.
>
>
>
> ## Q4: Minors
>
> A4: Thank you for finding the missing caption. We will update our revision.
>
> [4] Dataset Distillation, in arXiv 2018

---

### Official Review · Reviewer_2Cdm · 2025-10-29

**Soundness:** 2
**Presentation:** 3
**Contribution:** 2
**Rating:** 4
**Confidence:** 4

**Summary:**

This paper introduces a new evaluation benchmark for dataset distillation (DD) in image classification, aiming to assess the effectiveness of distilled datasets compared to random selection. The benchmark focuses on two proposed metrics: label-robust score and augmentation-robust score.
As summarized in Table 1, existing DD methods differ significantly in their use of labels (hard vs. soft labels, and whether soft labels come from a fully-trained teacher or are jointly optimized during distillation) and augmentations (e.g., resize-crop, patch-shuffle, cutmix). These differences make direct comparison of DD methods difficult. The paper argues that prior evaluations, which each use their own label and augmentation setups, are unfair and inconsistent.
To address this, the authors propose:
* Label-robust score: compares the accuracy of distilled data versus random selection under the same label setting (e.g., both using hard labels or the same soft labels).
* Augmentation-robust score: compares distilled data versus random selection under the same augmentation setting (e.g., same augmentation type or no augmentation).

The proposed benchmark aims to standardize evaluation conditions and reveal the true contribution of the distilled images themselves, separate from the effects of labels or augmentations.

**Strengths:**

* The paper provides a meaningful attempt to standardize the evaluation of dataset distillation methods, enabling a more controlled comparison against random selection under matched label and augmentation setups.
* The results highlight interesting findings: under hard-label usage, matching-based DD methods remain stronger than recent soft-label–based approaches, suggesting that much of the improvement in newer methods (e.g., SRe2L) may stem from knowledge distillation rather than from the intrinsic quality of the synthetic images.

**Weaknesses:**

* Limited applicability of the metric: Although the proposed metrics allow comparisons under matched label/augmentation setups, they do not measure the ultimate achievable performance of each DD method under its best hyperparameter and setup choices. Since DD performance also depends on factors like architecture, optimizer, and training configuration, comparing distilled datasets only under uniform conditions offers limited insight into each method’s full potential.
* Ambiguous interpretability of the two measures: The two metrics—label-robust and augmentation-robust scores—merely quantify relative test accuracies rather than any intrinsic quality of the synthetic datasets. It is unclear how these two scores should be used jointly or whether they could be unified into a single, more interpretable evaluation measure.
* Limited scope beyond image classification: The paper focuses solely on image classification. Modern distillation applications extend to vision-language and language model distillation, where data efficiency is more critical. It is unclear how the proposed robustness metrics could generalize to multimodal or text-based distillation tasks, limiting the broader applicability and fundamental impact of the proposed benchmark.

**Questions:**

1. On metric applicability:
    * How do the proposed label-robust and augmentation-robust scores reflect the best achievable performance of each DD method?
    * Could the benchmark be extended to allow comparisons when each method is evaluated under its own optimal settings (e.g., best label/augmentation choices)?
2. On metric design and coherence:
    * How should users interpret the two robustness scores jointly?
    * Is there a principled way to combine the label-robust and augmentation-robust scores into a single unified measure that better reflects dataset quality?
3. On generalization beyond image classification:
    * Can the proposed evaluation framework be adapted for multimodal or language model distillation tasks, where label and augmentation definitions are more complex?
    * If not, how might the authors envision extending these metrics to broader domains?

---

> ### Author Response · Authors · 2025-11-19
> **Rebuttal First Page**
>
> **W1.1&Q1.1:** How do the proposed label-robust and augmentation-robust scores reflect the best achievable performance of each DD method?
>
> **R1.1:** Thanks for the question. In DD-Ranking, although we **don't explicitly report** the "highest achievable performance of each DD method under its best hyperparameter and setup choices", **this is still evaluated** by $acc_{syn-any}$. However, we want to again emphasize that the performance of a DD method shouldn't depend on any data-irrelevant factor, such as architecture, optimizer, and training configuration. To fairly compare two methods, all data-irrelevant factors should be disentangled (please refer to the global comment for our simple proof of why DD-Ranking can fairly compare different methods). This is because with misaligned evaluation settings, the absolute values of accuracy are no longer reliable. This is perfectly demonstrated by our benchmark results that many soft-label methods fail to match random selection even though they continuously improve the previous "SOTA". However, we think that the reviewer's suggestion to also report each method's original performances is fairly reasonable. We will adjust our public leaderboard in the future to include this part, **only for reference, not for ranking**. Thanks again for the constructive suggestion.
>
> ---
>
> **W1.2&Q1.2:** Could the benchmark be extended to allow comparisons when each method is evaluated under its own optimal settings (e.g., best label/augmentation choices)?
>
> **R1.2:** Thanks for the question. As explained in R1.1, DD-Ranking in fact does compute each method's optimal performance. We didn't explicitly report them to avoid confusing readers. Also, in our Python package to release, we have already implemented the function to return and save the original/optimal. This is in fact another advantage of DD-Ranking that we provide a set of unified evaluation code for the DD community which accommodates most of existing evaluation settings.
>
> ---
>
> **W2.1&Q2.1:** How should users interpret the two robustness scores jointly?
>
> **R2.1:** Thanks for the question. Sorry that we cause confusion. In DD-Ranking, we are not combining these two scores to provide the ranking. Instead, each of the two scores is computed to evaluate a specific misalignment for current DD evaluation. Based on our research, label type and data augmentation are the two most misaligned settings that cause significant performance fluctuation. Therefore, we propose LRS and ARS to disentangle these two factors and fairly compare different methods. We believe a strong DD baseline should be both label-robust and augmentation-robust, meaning both scores are expected to be high.
>
> ---
>
> **W2.2&Q2.2:** Is there a principled way to combine the label-robust and augmentation-robust scores into a single unified measure that better reflects dataset quality?
>
> **R2.2:** Thanks for the question. Since LRS and ARS each evaluate a different aspect, we think a stronger baseline should have both high LRS and ARS (analogous to benchmarks in other domains, such as video generation which considers quality and semantics separately). Thus, we can simply take an average of the two to further rank different methods. However, the reason why we don't combine them is two fold. Firstly, we spare room for future methods to enhance a DD method from one specific aspect, such as being more label-robust or more augmentation-robust. We believe such work is also very important and impactful. Secondly, as we continue to maintain DD-Ranking in a long term, we acknowledge the possibility for any new misalignment to arise. In that case, we will update our benchmark to stay comprehensive.

---

> ### Author Response · Authors · 2025-11-19
> **Rebuttal Continued**
>
> **W3&Q3:**  Can the proposed evaluation framework be adapted for multimodal or language model distillation tasks, where label and augmentation definitions are more complex? If not, how might the authors envision extending these metrics to broader domains?
>
> **R3:** Thanks for the question. We appreciate the reviewer's concern on other modalities. Indeed, as mentioned in our future work, one important direction for DD-Ranking is to integrate DD for other modalities, though they have been under-explored at this point of time. We acknowledge the trend of developing multimodal dataset distillation systems ([1-4]) and will add relevant discussion into the revision. Since the core of DD-Ranking is fair and unified evaluation, it can be easily extended to datasets of other modalities. Our vision for multimodal DD evaluation aligns well with the DD-Ranking framework as follows:
>
> - From the "what-to-do" perspective, DD-Ranking addresses the unfairness in current DD evaluation. This applies to other modalities as well—evaluation is fair only when data-irrelevant settings are aligned across methods.
> - From the "how-to-do" perspective, DD-Ranking disentangles data-irrelevant factors (such as knowledge distillation and data augmentation) by comparing different methods' relative improvements against a random selection baseline. This can be easily adapted to other modalities since random selection always serves as a universal lower bound.
>
> As stated, DD-Ranking will be maintained long-term. We will track the latest progress in dataset distillation across all areas and enrich our benchmark to be more comprehensive.
>
>
>
> [1] Efficient Multimodal Dataset Distillation via Generative Models, NeurIPS 2025
>
> [2] Beyond Modality Collapse: Representations Blending for Multimodal Dataset Distillation, NeurIPS 2025
>
> [3] Dataset Distillation of 3D Point Clouds via Distribution Matching, NeurIPS 2025
>
> [4] Vision-Language Dataset Distillation, TMLR 2024

---

> ### Author Response · Authors · 2025-11-24
> **A summary of rebuttal for your convenience. We appreciate your feedback!**
>
> We thank you again for providing valuable review comments. To save your time, we summarize our response to your questions as follows:
>
> 1. **How to reflect best achievable performance:** Although the best achievable performance is not reported in our submission, we have implemented our tool package (to be released) that returns the best achievable performance to users. However, we want to emphasize that the best achievable performance is actually misleading, as it doesn't reveal the true informativeness of the synthetic dataset. Therefore, we highly recommend using the best achievable performance only for reference, not for comparison.
> 2. **Does DD-Ranking allow each method to use its own setting:** Yes, this is, in fact, one advantage of DD-Ranking. We provided a unified benchmark that includes most of the DD methods' evaluation settings. And in fact, the $ Acc_{sun\\_any}$ is obtained under the method's optimal setting.
> 3. **How should users interpret the two robustness scores jointly:** Thanks for the great question. The two metrics are designed to solve different unfairness issues; thus, we report them separately. Both metrics are positively correlated with the synthetic data informativeness.
> 4. **Can we combine LRS and ARS?** Yes, they can be combined into one metric, such as taking the average. However, since they measure two different aspects, we separate them mainly to allow future methods to study each aspect individually. For example, a method that is highly label-robust (i.e., works well under both hard and soft labels) is definitely a breakthrough in DD, even if it doesn't touch the augmentation part.
> 5. **Multimodal dataset distillation:** Thanks for the question. DD-Ranking is built on image dataset distillation mainly due to the fact that DD for other modalities is under-explored. However, we are aware of recent progress and will actively integrate multimodal dataset distillation into DD-Ranking. This is a very important future step for us to make sure DD-Ranking can be used as a standardized benchmark for all future methods. DD-Ranking can be easily applied to MDD since we can compare different methods fairly using a random selection baseline to disentangle data-irrelevant factors. However, we acknowledge that other modalities may have different label types or augmentation techniques. Therefore, we will keep track of the progress of MDD and release the DD-Ranking with MDD as soon as possible.
>
> If you have any further concerns, don't hesitate to let us know. We are looking forward to your reply!

---

### Official Review · Reviewer_KAsP · 2025-10-29

**Soundness:** 3
**Presentation:** 3
**Contribution:** 2
**Rating:** 6
**Confidence:** 4

**Summary:**

This paper identifies significant unfairness in current dataset distillation evaluation practices, mainly caused by inconsistent training configurations. In particular, the use of soft labels often leads to performance gains that originate from knowledge distillation rather than the actual quality of the synthetic data, while improvements from data augmentation do not necessarily indicate better dataset informativeness. To address these issues, the authors introduce three new evaluation metrics, namely Hard Label Recovery (HLR), Improvement Over Random (IOR), and Label Robust Score (LRS), which aim to disentangle the effects of knowledge distillation and data augmentation from the intrinsic performance of distilled datasets.

**Strengths:**

1. The paper clearly demonstrates that performance improvements in existing dataset distillation methods often result from knowledge distillation or data augmentation rather than from the informativeness of the synthetic images.
2. The proposed evaluation metrics for comparing the performance of different models are clearly defined.
3. The authors conduct experiments with LRS and ARS across different model architectures, teacher models, and hyperparameter settings to verify the robustness of the proposed method.

**Weaknesses:**

1. The paper spends excessive space analyzing the limitations of existing methods. This part is repetitive and should be condensed into a shorter empirical motivation section.
2. Although LRS and ARS are intuitively motivated, their theoretical foundation is weak and lacks conceptual depth.
3. In Section 3, the definition of DD RANKING is unclear. It is not specified whether it refers to LRS, ARS, or a combination of both.
4. In line 240, the description of the normalization method for computing ARS is vague and should be explicitly defined.

**Questions:**

In line 210, does the random subset refer to real images from the original dataset or to synthetic noise samples?

---

> ### Author Response · Authors · 2025-11-19
>
> **W1:** The paper spends excessive space analyzing the limitations of existing methods. This part is repetitive and should be condensed into a shorter empirical motivation section.
>
> **R1:** We thank reviewer for the constructive suggestion. We will trim down the motivation section and make the limitation part concise in the revision.
>
> ---
>
> **W2:** Although LRS and ARS are intuitively motivated, their theoretical foundation is weak and lacks conceptual depth.
>
> **R2:** Thanks for raising this concern. Although the evaluation of dataset distillation is always empirical, i.e. using the downstream task performance, we agree with the reviewer that some theoretical foundation is needed for DD-Ranking to be well-established. Here, we provide a theoretical analysis of why LRS is fair for evaluating two methods A and B with different settings. ARS can be derived similarly.
>
> **Definition of Fair Comparison:** A comparison between methods A and B is fair if and only if the evaluation metric isolates the **data quality** contribution while neutralizing **setting-dependent** performance gains. Formally, we let:
>
> - $ q_A, q_B $: **Intrinsic data quality** of methods A and B
> - $ \beta(\pi) $: performance boost from the evaluation setting $\pi $ (independent of data quality)
>
> Consider two DD methods:
>
> - Method A: distills $ \mathcal{D}_A $ with evaluation protocol $ \pi_A $ (e.g., hard labels, batch size 256, step learning rate scheduler)
> - Method B: distills $ \mathcal{D}_B $ with evaluation protocol $ \pi_B $ (e.g., soft labels, batch size 1024, cosine learning rate scheduler)
>
> We can decompose the performance (accuracy in the image classification task) as follows:
>
> - $ Acc(D_A,\pi_A)= q_A+ \beta(π_A)+ \epsilon_A $
> - $ Acc(D_B,\pi_B)= q_B + \beta(π_B)+ \epsilon_B $
>
> where $\beta(\pi)$ represents the performance gain brought by the evaluation setting $\pi$ and $\epsilon \sim N(0,1)$ is the residual term.
>
> From this, we can easily see that comparing accuracy directly is unfair since we have no clue about the relationship between $\beta(\pi_A)$ and $\beta(\pi_B)$.
>
> Now, we consider the random selection baseline: $Acc(D_{rand}, \pi_{rand}) = q_{rand} + \beta(\pi_{rand}) + \epsilon_{rand}$. We only show why IOR is fair and HLR can be derived similarly.
>
> - $IOR_{A} = Acc(D_A, \pi_A) - Acc(D_{rand_A}, \pi_{rand_A}) = q_A - q_{rand_A} + \beta(\pi_A) - \beta(\pi_{rand_A}) + \epsilon_A - \epsilon_{rand_A} = q_A - q_{rand_A} + \epsilon_A - \epsilon_{rand_A}$
> - $IOR_{B} = Acc(D_B, \pi_B) - Acc(D_{rand_B}, \pi_{rand_B}) = q_B - q_{rand_B} + \beta(\pi_B) - \beta(\pi_{rand_B}) + \epsilon_B - \epsilon_{rand_B} = q_B - q_{rand_B} + \epsilon_B - \epsilon_{rand_B}$
>
> We can cancel the $\beta$ terms because $\pi_A = \pi_{rand_A}$ and $\pi_B = \pi_{rand_B} $. Taking the expectation of the difference we have
> $$
> \mathbb{E}[IOR_A - IOR_B] = \mathbb{E}[q_A - q_B] - \mathbb{E}[q_{rand_A} - q_{rand_B}] + \mathbb{E}[\epsilon_A - \epsilon_B] - \mathbb{E}[\epsilon_{rand_A} - \epsilon_{rand_B}]
> $$
> Now, for the same original dataset $D$, the expected intrinsic quality of random selection should be a constant, meaning $\mathbb{E}[q_{rand_A} - q_{rand_B}] = 0$. Thus, we have $\mathbb{E}[IOR_A - IOR_B] = \mathbb{E}[q_A - q_B]$. **This shows that the comparison of $IOR$ is a direct comparison of the intrinsic synthetic data quality without any data-irrelevant factors.**
>
> From the above analysis, we can see that the core of DD-Ranking is "aligning settings", which guarantees the fair comparison. We again thank the reviewer for the question, and we **have added the analysis above to the Section 3**.
>
> ---
>
> **W3:** In Section 3, the definition of DD RANKING is unclear. It is not specified whether it refers to LRS, ARS, or a combination of both.
>
> **R3:** We are very sorry for causing this confusion. Our initial plan is to propose a benchmark called DD-Ranking, which includes different metrics for fair evaluation (LRS and ARS for now). We understand that the word "ranking" may cause confusion. Here, the ranking refers to LRS and ARS separately, meaning each LRS and ARS has a ranking. We expect future methods also to report these two metrics separately. This will be more clear when our leaderboard is public.
>
> ---
>
> **W4:** In line 240, the description of the normalization method for computing ARS is vague and should be explicitly defined.
>
> **R4:** Thanks for the question. The normalization is defined as follows
> $ ARS_{norm} = 100 \\% \times (e^{\beta} - e^{-1}) / (e - e^{-1}) $. This is the same as LRS. We do normalization to normalize the range of both metrics.
>
> ---
>
> **Q1:** In line 210, does the random subset refer to real images from the original dataset or to synthetic noise samples?
>
> **A1:** Thanks for the question. The random subset refers to the real images from the original dataset. The size of this subset if kept the same as the IPC. For example, for IPC=50 on ImageNet-1K, the random subset is uniformly sampled with each class having 50 images.

---

> ### Author Response · Authors · 2025-11-24
> **A summary of rebuttal for your convenience. We appreciate your feedback!**
>
> We thank you again for providing valuable review comments. To save your time, we summarize our response to your questions as follows:
>
> 1. **Theoretical analysis:** We have provided a theoretical proof of why LRS and ARS are fair metrics to compare different methods. The proof has been updated in the revision Section 3.4. The TL;DR is that LRS and ARS can successfully remove the accuracy boost from the data-irrelevant evaluation setting.
> 2. **Definition of DD-Ranking:** We are sorry for the confusion. DD-Ranking serves as the name of our benchmark, which consists of two metrics, LRS and ARS, each for one aspect of current unfair DD evaluation.
> 3. **Normalization of ARS:** The normalization is done via $ ARS_{norm} = 100\\% \times (e^{\beta} - e^{-1}) / (e - e^{-1})$
> 4. **Random subset:** The random subset refers to randomly selected real images.
> 5. **Adjusting the length of motivation:** Thanks for the suggestion. We have updated a trimmed version of Section 2 in the revision.
>
> If you have any further concerns, don't hesitate to let us know. We are looking forward to your reply!

---

> ### Comment · Reviewer_KAsP · 2025-11-27
>
> Thank you for the detailed rebuttal and clarifications provided. All my concerns are addressed.

---

> ### Author Response · Authors · 2025-11-27
> **We appreciate your support!**
>
> We are very happy to address your concerns. If you have any further questions, we are more than willing to explain in detail.

---

### Official Review · Reviewer_XoEQ · 2025-10-30

**Soundness:** 3
**Presentation:** 3
**Contribution:** 2
**Rating:** 6
**Confidence:** 4

**Summary:**

This paper argues that prior work on dataset distillation has been evaluated on an unfair playing field, and that simple accuracy comparisons do not guarantee fair or consistent assessment. To address this, the authors introduce DD-Ranking, which proposes a unified evaluation framework comprising four components to compare diverse DD methods under equitable criteria. A key strength is that DD-Ranking aims to deliver consistent evaluation irrespective of model architecture, the presence or absence of soft-label optimization, and the specific data-augmentation settings.

**Strengths:**

* The paper is well-written with a clear, thorough, and concise introduction that effectively summarizes key points
* The authors demonstrated through extensive experiments that the proposed evaluation metrics are meaningful and effective.

**Weaknesses:**

* Discussion of limitations is lacking
* Theoretical background would be needed

**Questions:**

## Discussion of limitations is lacking
* This paper evaluates performance only on datasets and models designed for classification tasks. I am curious how the authors envision establishing fair evaluation protocols in the context of multi-modal dataset distillation (MDD), where tasks and modalities may differ substantially.

## Theoretical background would be needed
* The paper aims to mitigate the unfairness introduced by data augmentation through the Augmentation-Robust Score (ARS). However, γ is fixed to 0.5, equally weighting the accuracy gaps obtained under augmentation and non-augmentation settings.
Conceptually, if the goal is to isolate and remove the influence of augmentation, measuring only the non-augmented accuracy gap (i.e., $acc_{syn-naug} − acc_{rdm-naug}$) might be sufficient and more principled. Why is the augmented gap ($acc_{syn-aug} − acc_{rdm-aug}$) necessary to include in the computation?
* Moreover, have the authors examined which of the two gaps (augmented or non-augmented) plays a more significant role in determining distillation quality?

---

> ### Author Response · Authors · 2025-11-18
>
> **W1:** This paper evaluates performance only on datasets and models designed for classification tasks. I am curious how the authors envision establishing fair evaluation protocols in the context of multi-modal dataset distillation (MDD), where tasks and modalities may differ substantially.
>
> **R1:** Thank you for raising this important question. As we stated in the conclusion & future work section, one limitation of DD-Ranking is that we currently focus only on image distillation. Our main justification is that image distillation has been sufficiently explored in recent years, while distillation for other modalities remains under-explored. However, we definitely acknowledge recent progress on multi-modal dataset distillation, such as [1, 2, 3, 4] (we will add this discussion in the revision).
>
> Our vision for MDD evaluation aligns well with the DD-Ranking framework as follows:
>
> - From the "what-to-do" perspective, DD-Ranking addresses the **unfairness** in current DD evaluation. This applies to other modalities as well—evaluation is fair only when data-irrelevant settings are aligned across methods.
> - From the "how-to-do" perspective, DD-Ranking disentangles data-irrelevant factors (such as knowledge distillation and data augmentation) by comparing different methods' relative improvements against a random selection baseline. This can be easily adapted to other modalities since random selection always serves as a universal lower bound.
>
> As stated, DD-Ranking will be maintained long-term. We will track the latest progress in dataset distillation across all areas and enrich our benchmark to be more comprehensive.
>
> [1] Efficient Multimodal Dataset Distillation via Generative Models, NeurIPS 2025
>
> [2] Beyond Modality Collapse: Representations Blending for Multimodal Dataset Distillation, NeurIPS 2025
>
> [3] Dataset Distillation of 3D Point Clouds via Distribution Matching, NeurIPS 2025
>
> [4] Vision-Language Dataset Distillation, TMLR 2024
>
> ------
>
> **W2**: Conceptually, if the goal is to isolate and remove the influence of augmentation, measuring only the non-augmented accuracy gap might be sufficient and more principled. Why is the augmented gap necessary to include in the computation?
>
> **R2**: Thank you for this question. The augmented gap is included because we allow different methods to use augmentation, just as we allow soft labels. What DD-Ranking emphasizes is that many methods may be using these techniques incorrectly—they are only applied during test time and don't contribute to improving the synthetic data during distillation. However, we also acknowledge methods that successfully integrate data augmentation (e.g., DSA) or knowledge distillation (e.g., DATM) into the distillation process and truly improve data informativeness. Therefore, the augmented gap (and soft-labeling gap) is computed to analyze whether the use of augmentation (or soft labels) genuinely improves synthetic data quality, and by how much compared to the random selection baseline.
>
> ------
>
> **Q1**: Moreover, have the authors examined which of the two gaps (augmented or non-augmented) plays a more significant role in determining distillation quality?
>
> **A1**: Thank you for this excellent question. We interpret your question in two possible ways and address each below:
>
> - *For DD-Ranking, which gap dominates the ranking?*
>   Based on our evaluation of baselines on ImageNet-1K, the non-augmented gap is more important when IPC is large, while the augmented gap dominates when IPC is small. This is because when IPC is small (e.g., IPC=1), accuracies of both baseline methods and random selection are low without augmentation (~5%). However, as IPC increases, random selection performance becomes non-trivial, but many baseline methods struggle to perform comparably when data augmentation is removed (e.g., random resized cropping, which our experiments show is particularly effective for soft-label methods).
> - *Which gap better reveals the true synthetic data quality?*
>   This is an insightful question. The motivation for building DD-Ranking is to propose a fair and comprehensive framework for comparing different methods. Since synthetic data distilled by most methods are not human-interpretable, the most effective way to measure quality is through downstream task performance. We use both gaps to encourage future methods to leverage data augmentation to genuinely improve synthetic data quality. Thus, our answer is that both gaps are *equally important*. However, as shown in the submission, ARS is reported separately for w/ and w/o augmentation, as well as combined. This allows future exploration to study how to distill augmentation-robust datasets.
>
> **Please find our theoretical analysis of the fairness of DD-Ranking in the global comment.**

---

> ### Author Response · Authors · 2025-11-24
> **A summary of rebuttal for your convenience. We appreciate your feedback!**
>
> We thank you again for providing valuable review comments. To save your time, we summarize our response to your questions as follows:
>
> 1. **Discussion of Limitation:** DD-Ranking is built on image dataset distillation mainly due to the fact that DD for other modalities is under-explored. However, we are aware of recent progress and will actively integrate multimodal dataset distillation into DD-Ranking. This is a very important future step for us to make sure DD-Ranking can be used as a standardized benchmark for all future methods.
> 2. **How to apply DD-Ranking on MDD:** Fair evaluation, which is the main focus of DD-Ranking, should be the core for any modalities. In that sense, DD-Ranking can be easily applied to MDD since we can compare different methods fairly using a random selection baseline to disentangle data-irrelevant factors. However, we acknowledge that other modalities may have different label types or augmentation techniques. Therefore, we will keep track of the progress of MDD and release the DD-Ranking with MDD as soon as possible.
> 3. **Why compute two gaps for ARS:** Although we claim that soft labeling and data augmentation are the two major unfair tricks currently being used in evaluation, we do not prevent future methods from using these two tricks properly to truly improve the synthetic data quality. Therefore, we always keep the soft label gap and the augmentation gap to see how much improvement the technique can bring to the synthetic data.
> 4. **Which gap plays a more significant role:** For DD-Ranking, we treat both gaps equally important because they are complementary. ARS is reported separately for w/ and w/o augmentation, as well as combined. This allows future exploration to study how to distill augmentation-robust datasets.
> 5. **Theoretical foundation:** We have proposed a theoretical proof of why DD-Ranking is a fair benchmark. The proof can be found both in the global response and in the revision Section 3.4.
>
> If you have any further concerns, don't hesitate to let us know. We are looking forward to your reply!

---

### Author Response · Authors · 2025-11-19
**General Response**

We thank all reviewers for constructive feedback. Here, we provide the response to some commonly asked questions.

---

## Theoretical Foundation

We provide a theoretical analysis of why LRS is fair for evaluating two methods, A and B, with different settings. ARS can be derived similarly.

**Definition of Fair Comparison:** A comparison between methods A and B is fair if and only if the evaluation metric isolates the **data quality** contribution while neutralizing **setting-dependent** performance gains. Formally, we let:

- $ q_A, q_B $: **Intrinsic data quality** of methods A and B
- $ \beta(\pi) $: performance boost from the evaluation setting $\pi $ (independent of data quality)

Consider two DD methods:

- Method A: distills $ \mathcal{D}_A $ with evaluation protocol $ \pi_A $ (e.g., hard labels, batch size 256, step learning rate scheduler)
- Method B: distills $ \mathcal{D}_B $ with evaluation protocol $ \pi_B $ (e.g., soft labels, batch size 1024, cosine learning rate scheduler)

We can decompose the performance (accuracy in the image classification task) as follows:

- $ Acc(D_A,\pi_A)= q_A+ \beta(π_A)+ \epsilon_A $
- $ Acc(D_B,\pi_B)= q_B + \beta(π_B)+ \epsilon_B $

where $\beta(\pi)$ represents the performance gain brought by the evaluation setting $\pi$ and $\epsilon \sim N(0,1)$ is the residual term.

From this, we can easily see that comparing accuracy directly is unfair since we have no clue about the relationship between $\beta(\pi_A)$ and $\beta(\pi_B)$.

Now, we consider the random selection baseline: $Acc(D_{rand}, \pi_{rand}) = q_{rand} + \beta(\pi_{rand}) + \epsilon_{rand}$. We only show why IOR is fair, and HLR can be derived similarly.

- $IOR_{A} = Acc(D_A, \pi_A) - Acc(D_{rand_A}, \pi_{rand_A}) = q_A - q_{rand_A} + \beta(\pi_A) - \beta(\pi_{rand_A}) + \epsilon_A - \epsilon_{rand_A} = q_A - q_{rand_A} + \epsilon_A - \epsilon_{rand_A}$
- $IOR_{B} = Acc(D_B, \pi_B) - Acc(D_{rand_B}, \pi_{rand_B}) = q_B - q_{rand_B} + \beta(\pi_B) - \beta(\pi_{rand_B}) + \epsilon_B - \epsilon_{rand_B} = q_B - q_{rand_B} + \epsilon_B - \epsilon_{rand_B}$

We can cancel the $\beta$ terms because $\pi_A = \pi_{rand_A}$ and $\pi_B = \pi_{rand_B} $. Taking the expectation of the difference we have
$$
\mathbb{E}[IOR_A - IOR_B] = \mathbb{E}[q_A - q_B] - \mathbb{E}[q_{rand_A} - q_{rand_B}] + \mathbb{E}[\epsilon_A - \epsilon_B] - \mathbb{E}[\epsilon_{rand_A} - \epsilon_{rand_B}]
$$
Now, for the same original dataset $D$, the expected intrinsic quality of random selection should be a constant, meaning $\mathbb{E}[q_{rand_A} - q_{rand_B}] = 0$. Thus, we have $\mathbb{E}[IOR_A - IOR_B] = \mathbb{E}[q_A - q_B]$. **This shows that the comparison of $IOR$ is a direct comparison of the intrinsic synthetic data quality without any data-irrelevant factors.**

---

## Missing Results for Random Noise with Soft Labels

We are very sorry for the missing results. Here are the experiment results corresponding to Section 2.2.

**Experiment Settings**

- IPC: 50
- Model: ResNet-18
- Soft Label Temperature: 20.0 (same as SRe2L, RDED, etc.)
- Data Augmentation: RandomResizedCrop (same as SRe2L)
- Learning Rate: 1e-3
- \#Epochs: 300
- Optimizer & Scheduler: AdamW & CosineAnnealing

**Experiment Results**

| dataset    | random guessing | batch size=128 | batch size=1024 |
| :--------- | :-------------- | :------------- | :-------------- |
| ImageNet1K | 0.1%            | 0.5%           | 0.3%            |

As shown in the table, under different batch sizes, the pure noise outperforms random guessing. These results provide strong evidence for our claim that soft labels (used at the evaluation time of DD) don't contribute to data quality, but only improve the test accuracy.

---

## Revision Update

We have updated the revision following all reviewers' suggestions (text in red), including
- Table 2a and 2b: Missed results for random noise and added results for noisy images under soft labels
- Section 2: Trimmed version
- Line 163-164: Discussion and acknowledgement of related work
- Section 3.4: The theoretical analysis, same as above
- Line 508-511: Discussion of multimodal dataset distillation
- Appendix A: Discussion of different $ \lambda $ values.

---

### Meta-Review · Area_Chair_zxF4 · 2026-01-10

**Summary:**

The authors argue that recent approaches for dataset distillation/condensation have not been evaluated fairly, since they exhibit substantial methodological differences: some use soft labels and others do not, some use augmentation and others do not, some use multiple teachers and others do not, etc. To reconcile this they propose a new evaluation framework (DD-Ranking) and also propose new scores/metrics that plausibly can be used to compare methods in a fairer fashion. Experimental results on simple image classification benchmark datasets are provided in support of the approach.

The paper received borderline reviews. The proposed scores (LRS and ARS) seemed somewhat heuristic, and reviewers pointed out the lack of theoretical justification. A concern was also raised regarding the evaluation framework (which seemed to structurally disadvantage methods which relied on data augmentation). Concerns regarding extensibility of the approach beyond simple image classification were also raised. Apart from these there were writing issues pointed out in several places. The authors provided lengthy responses to these concerns.

Unfortunately, I don't think the responses were altogether very convincing. The authors provided theoretical justification, which to be honest appears to be very handwavy and not adding much to the manuscript. Above all, I echo the concern about disadvantaging augmentation techniques. I don't think the analogy to LLM prompt engineering given by the authors in the rebuttal necessarily rings true, and augmentation techniques such as CutMix are now widely accepted as standard ingredients in SOTA methods for ImageNet-1K.

Overall, this seems like a good (although somewhat limited) contribution to the field, but the unresolved concerns prevent me from recommending acceptance.

**Reviewer Concerns:**

The concerns about the theoretical justification, evaluation framework, and extensibility to tasks beyond image classification all remain unresolved.

**Reviewer Scores:**

The scores were 6,6,4,4. One of the '6' scores said that they were happy with the responses, and may have improved their score (although their comments were not super critical in the first place). I am not sure either of the '4' reviewers would have changed their score.

---

### Decision · Program_Chairs · 2026-01-26

Reject